# Optimizing rice-fish co-culture: Investigating the impact of rice spacing density on biochemical profiles and production of genetically modified tilapia (*Oreochromis spp*.) and *Cyprinus carpio*

**Muhammad Inayat**[1]*, **Farzana Abbas**[1], **Muhammad Hafeez-ur-Rehman**[1], **Athar Mahmud**[2]

1 Department of Fisheries and Aquaculture, University of Veterinary and Animal Sciences, Lahore, Pakistan,
2 Department of Poultry Production, University of Veterinary and Animal Sciences, Lahore, Pakistan

* muhammad.inayat@uvas.edu.pk

**Data Availability Statement:** All relevant data are within the paper and its Supporting Information files.

## Abstract

Rice fish co-culture synergistically boosts food production, resulting in numerous advantages across the environmental, social, and financial domains. A study was conducted to investigate the effects of three different rice spacing densities, rice high density (RHD) 9inch×12inch, rice medium density (RMD) 12inch × 12inch, rice low density (RLD) 15inch×12inchon both rice and fish. Various parameters were assessed to evaluate the performance of the co-culture system, including water quality, growth parameters, muscle quality, soil characteristics, rice stem characteristics, and rice yield parameters. When comparing the species, it was observed that GMT (Genetically Male Tilapia) demonstrated superior weight gain (303.13g vs 296.41g) and specific growth rate (1.16 vs 1.15). Regarding the proximate composition, results showed that RMD had the highest crude protein and fat content compared to RLD and RHD. GMT also exhibited greater crude protein and fat content than *Cyprinus carpio*, with RMD showing the highest values. Treatment groups significantly influenced the amino acid profile of experimental species, with RMD exhibiting the highest values. GMT showed significantly higher levels of essential, non-essential, half-essential, and umami amino acids compared to *Cyprinus carpio*. The interaction between RMD and GMT further demonstrated significant differences in various amino acid categories with RMD. A non-significant difference was observed among the treatments regarding soil biochemical characteristics. Regarding the rice stem characters, the height of the plant, panicle length, and stem length of rice were found to be comparable in the RMD and RLD groups however, significantly higher in RHD. Regarding rice yield parameters, no significant differences were observed among the other treatment groups, except for yield per hectare (yield/ha), which was significantly higher in the RHD group compared to RMD and RLD. Additionally, 1000-grain weight and panicle number (ears per hill) were significantly higher in the RLD treatment than in the other treatments. In conclusion, our findings indicate that the RMD treatment consistently yielded superior results compared to RLD and RHD. Furthermore, within the rice-fish co-culture system, GMT proved to be a more competent

**Funding:** The Punjab Agriculture Research Board (PARB) provided the funds to develop an Integrated Aquaculture Research Unit (IARU). The funders had a role in study design, but there was no role in data collection and analysis, the decision to publish, or the preparation of the manuscript.

**Competing interests:** The authors have declared that no competing interests exist.

species compared to *Cyprinus carpio*. The study provides data to understand the interactions between rice spacing density, fish growth and overall productivity can guide the development of sustainable and profitable rice-fish co-culture systems.

## Introduction

Rapid population growth poses many problems, specifically malnutrition, shortage of food, and limited land and water resources that are directly related to agricultural production and environmental degradation. The most serious problems that are considered challenges for developing countries are hunger, malnutrition, and poverty [1]. Worldwide, rice is a major crop and is mostly cultured in Asia [2], is considered for worldwide food security [3]. The production of *Oryza sativa L*. (rice), which consumes up to 90% of Asia's irrigated water, covers almost 155m ha of land and provides approximately 50% of the total population of the world's rice production [4, 5]. Sustainable production of rice is increased by the new applied techniques in agricultural systems. Food quality and quantity increase reduced the application of pesticides and fertilizers by conducting scientific research under different sustainable and economic conditions [6].

In Pakistan, wheat and rice hold supreme significance in the agricultural sector, serving as the primary staple foods. Pakistan ranks among the world's top rice producers, with rice being the second most consumed staple. Rice is a widely cultivated crop across all provinces, ranging from latitude 24˚ to 36˚ and spanning diverse environmental conditions, from sea level in the south to high altitudes of 2500 meters in the northern mountains. These regions encompass coastal tropical humid and arid hot plains areas. In Punjab, rice cultivation grows well in districts like Kasur, Hafizabad, Nankana Sahib, Narowal, Gujranwala, Sheikhupura, Sialkot, Chiniot, Gujrat, and Mandi Bahauddin. In Sindh, it is prominent in Badin, Thatta, Jacobabad, Larkana, Shikarpur, Dadu, & extends to Nasirabad and Jaffarabad districts in Baluchistan [7]. Agriculture serves as the predominant livelihood in Kasur, with the region specializing in the cultivation of various crops such as wheat, rice, sugarcane, and cotton, as well as a variety of fruits and vegetables. Changa Manga, established in 1864 as a forestry plantation, has since evolved to become a diverse source of products including silk, honey, beeswax, turmeric, and high-quality timber [8].

The integration of aquatic animals with rice (e.g., ducks, fish, mussels, shrimp, and crab) is one of the most sustainable production systems by reducing ecological costs. The combination of agriculture with aquaculture is known as the integrated agri-aquaculture farming system (IAAFS). The IAAFS efficiently endorses sustainable agri-aquaculture development [9]. The culturing of loach, tilapia, catfish, shrimp, and carp are most suitable in rice paddies [10]. The system can also decrease environmental pollution, improve water and land resource usage, and provide grain and meat for consumers [11]. In recent decades, aquaculture sector has received a lot of attention due to the maximum production of aquatic food that sustains individual consumption. In the diet of developing countries, freshwater aquaculture plays an important role in providing animal-based protein, fatty acids, vitamins, amino acids, and vitamins [12]. Combined farming has great prospective to emerge as an effective tool aimed at the development of the rural economy due to high profitability and low investment [13]. The co-culture system has been observed to reduce pest communities and lower the requirement for the agro-chemical usage including pesticides and herbicides. Several studies [14, 15] reported that rice-fish generated extraordinary aquaculture production, which increased farmers'

incomes. The fish consume the pests, and insects, while also providing organic fertilizer through their excrement. This reduces the need for chemical pesticides and fertilizers, making the farming process more environmentally friendly. In Asian countries, the co-culture of rice-fish has been practiced in the field of rice for more than 2000 years e.g., Vietnam [3], China [16], Bangladesh [14] and Malaysia [16]. However, in Pakistan, the integrated farming system for the development of organic products is still silent. Concerns are related to the safety and quality of products or various other issues related to environments such as low energy production, consumption of large volumes of water, and untreated wastewater [17].

Given the current economic needs in Pakistan, the imperative is for farmers to adopt an outcome-based farming system. This approach ensures not only economic prosperity for farmers but also reinforces and sustains food security. Integrated fish farming offers hope in this direction as it serves as a food-production base that combines the farming of crops, rearing of livestock/poultry, and fish farming. The increasing population growth and its demand for food are pushing worldwide agriculture production [18]. Hence, the agricultural sector faces the imperative of exploring and implementing sustainable technologies to address the tripartite challenges of intensification, expansion, and climate change. This necessitates a comprehensive approach that integrates innovative technologies to enhance productivity, expands cultivation practices responsibly, and fortifies resilience against the impacts of a changing climate. The example of rice production is ecologically intensified by alternating and sharing land usage by developing rice-fish co-culture systems [19]. This system eventually reduced the damage to environmental health and indirectly improved farm productivity and income. In addition, future technological methods of production of a crop may also implement a combined farming system where the potential for declining ecosystems and the effects of global warming are greatly reduced and decrease the operational cost in agriculture, which might serve as a potential for future implementation of integrated farming systems [20].

Plant spacing has a large effect on plant growth: a very high density can lead to extreme competition for soil nutrients between neighboring crops, while excessive spacing is a waste of growing areas [21]. The density of rice spacing can affect the rice yield by altering the number of panicles per unit area of land and the feeding rate can affect the fish growth. A previous study showed that the 30cm×30cm rice spacing density reduced the yield of rice but increased the growth of fish [22]. The integration of rice-fish systems has been widely practiced in Asia but is still relatively underutilized in other regions. Furthermore, despite the promising outcomes of previous studies, the ideal density of rice and fish species for maximum yield, productivity, and profitability has yet to be determined.

In recent years, the study of rice-fish culture systems has become increasingly significant for sustainable agriculture. This research aims to fill a knowledge gap by investigating the optimized integration of rice and fish cultivation. Specifically, hypothesize that optimizing the integration, particularly through optimal plant spacing; will enhance the growth of both components. The study aims to evaluate the integration of rice and fish cultivation, as well as the effect of different rice plant spacing on the growth of both rice and fish, and any other ecological factors.

## Materials and methods

### Description of field

The experiment was conducted in the Integrated Aquaculture Research Unit (IARU) at Department of Fisheries and Aquaculture, University of Veterinary and Animal Sciences, Ravi Campus, Pattoki, Kasur, Pakistan (31˚1'0 N 73˚50'60 E with an altitude of 186 meters). The study area map is presented in Fig 1. Pattoki is located in Punjab, a typical subtropical

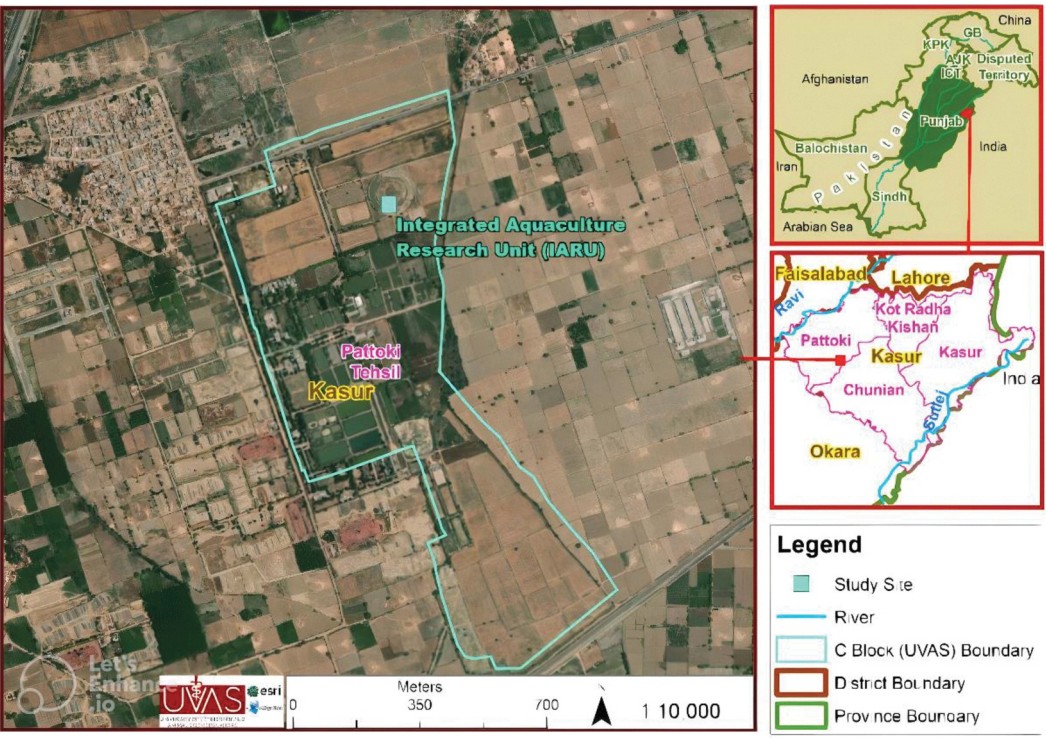

**Fig 1. Study area map developed by the Remote Sensing and GIS Laboratory, Department of Wildlife and Ecology, UVAS, Ravi campus, Lahore, through the software ArcGIS®10.5.1.**

monsoon climate zone. The maximum elevation above the level of the sea is 199 meters. Summers are humid, clear, and warm and winters are dry, very clear and short. The Average annual precipitation is 500 mm. Over the course of a year, the temperature fluctuates from 45˚F to 104˚F and is hardly under 41˚F or above 111˚F. The soil of the experimental site is loamy containing, 5.8 pH, organic matter (gkg-1) 12.6, 18.7 (mgkg-1) available nitrogen, total nitrogen (mgkg-1) 0.14, available phosphorous 6.8 (mg kg-1), available potassium 69.9 (mg kg-1.).

## Experimental design

Ethical approval for conducting this study was granted by the animal use and animal care committee of the University of Veterinary and Animal Sciences, Lahore, confirming compliance with established standards of animal welfare. For the present assessment, 12 experimental plots with a rice-fish co-culture system were selected randomly. Three different types of rice planting spacing (we called it rice spacing density) were maintained as treatment and each treatment was replicated in triplicate. The details of these treatments are presented in (Fig 2).

  T1 = RHD (Rice-high-density)
  T2 = RMD (Rice-medium-density)
  T3 = RLD (Rice-low-density)

In the experimental field of rice, opposite corners are opened, inlets and outlets respectively. The size of each plot is 6500 (Sq ft). The rice was irrigated and the water level was maintained to a specific level suitable for fish culture (15cm to 20 cm). Anti-birds net was used. Before rice planting, as a base fertilizer duck manure was used in rice plots, and no top dressing was applied. The rice variety was Super kainat No. 1121 and two fish species Genetically Male

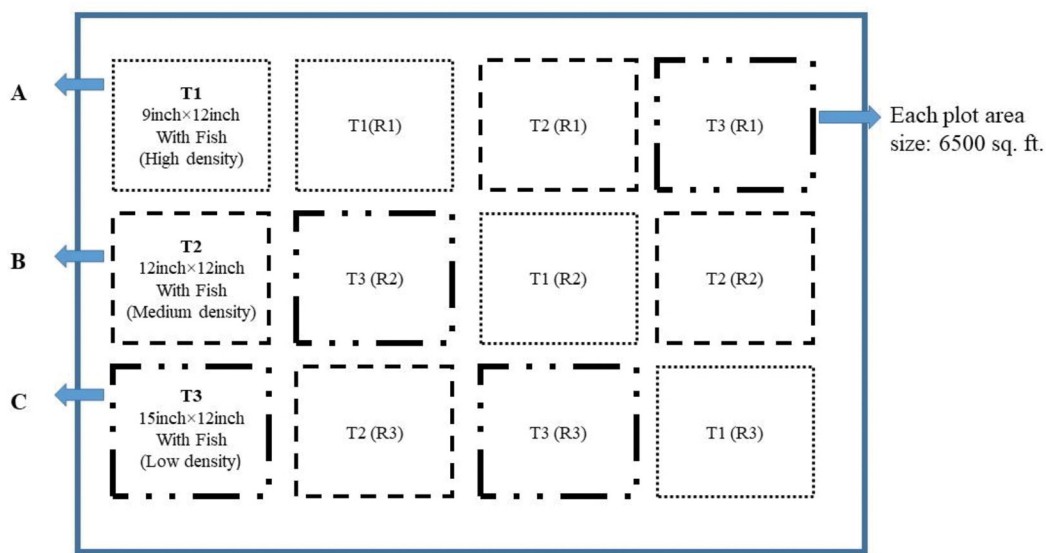

**Fig 2.** Experimental design considering 12 plots, A = Treatment 1 (High density, RHD), B = Treatment 2 (Medium density, RMD) and C = Treatment 3 (Low density, RLD).

Tilapia (GMT) and *Cyprinus carpio* were used for this experiment. Rice was planted on June 5, 2020, and harvested on November 15, 2020. The fishes were procured from Chenab Fish Hatchery, Rangpur, Punjab. Fish acclimatization is a crucial step in the successful implementation of rice-fish co-culture systems. Common carp and genetically male tilapia (GMT) were used in this study. The optimal conditions were used for the acclimatization phase in order to ensure the fish's growth and survival in the rice field. The common carp were acclimated to a water temperature range of 20–28˚C, a water pH range of 6.5–8.5, a dissolved oxygen (DO) level of at least 5 mg/L, a light intensity of 0.5–1.5 watts/m$^2$, and a diet that included plant and animal-based protein. They were quarantined for 14 days before released into the rice field. Similarly, GMT was acclimated to a water temperature range of 28–32˚C, a water pH range of 6.5–8.5, a dissolved oxygen (DO) level of at least 5 mg/L, a light intensity of 0.5–1.5 watts/m$^2$, and a diet that included plant and animal-based protein. Before being released into the rice field, the fish underwent a 14-day quarantine period. The acclimatization process was carried out gradually over several days, with meticulous monitoring to ensure the welfare of the fish and the successful establishment of the rice-fish co-culture system. After 20 days of rice transplantation, both fishes were stocked (9000/ha), with an average weight of 50±3.0 g in a 1:1 ration to each plot. Fishes were fed on commercial feed (Supreme Aqua feed; 25% CP) 2% body weight twice a day and purchased from Supreme Feeds (Private limited). According to the condition of feeding and growth, the quantity of feed was simultaneously increased.

## Sample collection and processing

Eight days before the rice harvest, the fish were caught from each plots, weighed, and counted. 72 fishes were randomly selected for meat analysis (Proximate analysis = 36 and amino acid profile = 36). After the collection of fishes, scales were removed, skin, and intermuscular thorns. Fish muscles were cut and merged into a sample; these samples were carefully packaged in labeled, airtight plastic bags and stored at -20˚C for further analysis. Phytoplankton and zooplankton were sampled fortnightly using a plankton net. Following to collection, a rigorous analysis was conducted to ascertain and quantify the diversity of both phytoplankton

and zooplankton populations in the study environment [23]. Water sample was collected in the three different corner of paddy field for the identification of plankton. In order to collect Zooplankton samples, 20 liters of mixed water were filtered through a 112-cm plankton net and stored in formaldehyde at 4%. A 1 L mixture of mixed water was withdrawn and an iodine solution was added to collect phytoplankton samples. A laboratory analysis was performed in order to identify the species and determine the biomass volume of phytoplankton and zooplankton. Microscopes (Olympus CX31) and stereoscopes (Olympus SZX10, Japan) were used and with the help of available keys and manual, plankton grouping were identified [24, 25]. Additionally, rice samples were collected from each plot to calculate key parameters for determining the yield of the rice crop.

## Growth parameters

Before the experiment, Fish live weight was recorded fortnightly and final measurements were taken at the end of the experiment. The growth parameters like weight gain, and specific growth rate (SGR) were calculated by following the equation [26].

Weight gain = Final body weight (g)–initial body weight (g)

SGR = [(Final weight–initial weight)/days of growth trials)] × 100

## Proximate analysis

The proximate analysis of fish samples (three fish per plot) was evaluated by standard method [27]. The soxhlet extractor method used to identify the content of crude fat [28]. Crude protein was evaluated by the Kjeldahl method [29]. The moisture content was measured by using a natural convection furnace at 105˚C until a constant weight was reached and a muffle furnace at 550˚C determined the content of ash for 4 hours.

Crude Fat % = (Weight of Fat in the Fish / Total Weight of the Fish) x 100

Crude Protein (in %) = (N x 6.25) / W

Where:

N: Nitrogen content determined from the sample.

6. 25: Conversion factor, assuming that proteins contain approximately 16% nitrogen.

W: The weight of the sample in grams.

## Evaluation of amino acid profile

The amino acid profile of fish muscle samples (three fishes per plot) was calculated using an amino acid analyzer (Biochrom 30+, Biochrom Limited, USA) following the protocol described by [30]. The samples were thoroughly crushed until they reached a particle size of five hundred microns. To preserve cysteine and methionine, the samples were oxidized with formic acid. During this oxidation process, methionine was changed to methionine sulfone, and cysteine into cysteic acid. After oxidation, the fish samples were hydrolyzed by using a solution of six molar hydrochloric acid and phenol for a duration of twenty-four hours. The pH value of the samples was then adjusted to 2.2. After filtration, the samples were carefully transferred into small sample bottles in preparation for the quantification of amino acids. This quantification was performed using an ion-exchange chromatography method with the utilization of a Biochrom 30+ amino acid analyzer.

## Yield parameter of rice

The rice yield parameters were measured by following formulas [31]:

1. Seed setting rate (%) = filled grain number per panicle /Total grain number per spike × 100%

2. Panicle setting rate (%) = the number of effective spikes/numbers of highest seedling spikes ×100%

3. Grain No. per rice spike (Grain/ear) = total 5-hill grains number /total 5-hill spikes number

4. Number of hills per hectare (hills/ha) = $\dfrac{10000}{\text{(Distance (m) between rows} \times \text{distance (m) between hills)}}$

5. Yield (kg/ha) = 20 hill rice weight (kg)/20 × Hill number per hectare

## Monitoring of water quality parameter

Water quality parameters, including Total Dissolved Solids (TDS), temperature (Temp), Electrical Conductivity (EC), pH, and Dissolved Oxygen (DO), were monitored twice daily(Morning and Evening) throughout the experimental duration by using HANNA HI 98194 multimeter.

## Soil bio-chemical characterizations

Depending on the rice spacing, a comparison of the soil biochemical profile was done. Ten soil cores (0–20 cm) distributed randomly in each experimental plot were collected using a soil borer and assembled. Each soil sample was air-dried and filtered through Whatmann (2 and 0.15) filter paper analyses. The soil analysis was computed for potassium, nitrogen and phosphorus by method. To estimate the nitrogen content in the soil, the samples of soil were filtered and subjected to the Kjeldahl apparatus. Soil pH was measured by using pH meter BPPH-60 [32].

## Statistical analysis

Effects of different rice densities on growth performance, meat proximate, and amino acid profile of two fish species were evaluated through factorial ANOVA using PROC GLM in SAS software (version 9.1). Rice densities and fish species were considered as the main effects and their interactions were tested too. The comparison of significant treatment means was calculated through Duncan's Multiple Range test considering $p \leq 0.05$ (Model 1).

The data related to soil and rice stem characteristics, along with rice yield parameters, were subjected to analysis using a one-way ANOVA technique via the GLM procedure in SAS. Post hoc comparisons of significant treatment means were performed using Fisher's Least Significant Difference (LSD) test, with a significance level of $p < 0.05$ (Model 2). $\mathbf{Y_{ijk} = \mu + \alpha_i + \beta_j + (\alpha\beta)_{ij} + \epsilon_{ijk}}$ (Model 1)

Where,

$Y_{ijk}$ = Observation of dependent variable recorded on $i^{th}$ and $j^{th}$ treatment groups

$\mu$ = Population mean

$\alpha_i$ = Effect of $i^{th}$ treatment group (i = 1,2,3)

$\beta_j$ = Effect of $j^{th}$ treatment group (i = 1,2)

$(\alpha\beta)_{ij}$ = Interaction effect of $i^{th}$ and $j^{th}$ treatment groups

$\epsilon_{ijk}$ = Residual effect of $k^{th}$ observation on $i^{th}$ and $j^{th}$ treatment group, NID ~ 0, $\sigma^2$

$Y_{ij} = \mu + \tau_i + \epsilon_{ij}$ (Model 2)

Where,

$Y_{ij}$ = Observation of dependent variable recorded on $i^{th}$ treatment group

μ = Population mean

$\tau_i$ = Effect of $i^{th}$ treatment group (i = 1,2,3)

$\epsilon_{ij}$ = Residual effect of $j^{th}$ observation on $i^{th}$ treatment group, NID ~ 0, $\sigma^2$

## Results

### Growth performance

Before the harvesting of rice, the weight gain, and specific growth rate of fishes were calculated (Fig 3). The results indicated that the overall growth rates were increased with the increase of rice planting density showing significant results among all treatments. Regarding species, GMT showed the highest body weight (weight gain) and SGR as compared to *Cyprinus carpio* (303.13 vs. 296.41g; P = 0.0298). In terms of different treatment groups, RMD had the highest weight gain of fish followed by RLD and RHD, (334.55 vs. 294.35 vs. 270.41; P < 0.0001). The interaction between species and treatment showed significant differences. Interaction of GMT and *Cyprinus carpio* with RMD, it revealed the highest weight gain and SGR (P<0.0001).

### Proximate analysis

Under different rice spacing densities, least square mean ± standard errors of crude protein and fat of the muscles of the fish are present in (Fig 3). There were several differences regarding crude protein and fat among different treatments groups. In RMD, the content of crude protein was better as compared to other treatment groups. The fat contents showed variable results among different treatment groups. In terms of species comparison, GMT showed the highest crude protein and fat content in different treatments as compared to *Cyprinus carpio* (32.02 vs 23.52[b]; < 0.0001 and 10.08[a] vs 3.77[b]; P < 0.0001). Regarding treatment groups, medium rice spacing density (RMD) reached the highest value as compared to low (RLD) and high rice spacing density (RHD) for crude protein and fat (29.68 vs 26.93 vs 26.7; P = 0.0022 and 7.90 vs 6.48 vs 6.41; P = 0.0017). The interaction between species and treatments, there were significant differences in their crude protein and fat content RMD, GMT and *Cyprinus carpio* had the highest levels of crude protein (P<0.0001). GMT in RMD had the highest crude fat content, while *Cyprinus carpio* had the highest value in RLD (P<0.0001).

### Evaluation of amino acid

Least square mean ± standard errors of the Amino acid profile of experimental species are presented (Table 1). There were significant differences observed in the amino acid content among the different treatment groups. In species comparison, all essential amino acid is Threonine (Thr), Valine (Val), Methionine (Met), I Isoleucine (Ile), Leucine (Leu), Phenylalanine (Phe), Lysine (Lys), non- essential amino acid is Serine (Ser), Tyrosine (Tyr), Proline (Pro), half essential amino acid is Histidine (His), Arginine (Arg), and umami amino acid are Aspartic Acid (Asp), Glutamic acid (Glu), Glycine (Gly), Alanine (Ala) were found significantly higher (P < 0.0001) in GMT.

In treatment comparison, RMD (medium rice spacing density) showed the highest values as compared to other treatment groups. Regarding the interaction between treatments and species, RMD with GMT and *Cyrinus carpio* showed significant differences (P<0.0001) among essential, non-essential, half-essential, and umami amino acids.

### Rice stem characteristics

Before the harvesting of rice, rice stem characteristics are presented (Table 2). In RMD the height of the plant, panicle length, and stem length of rice were slightly higher as compared to

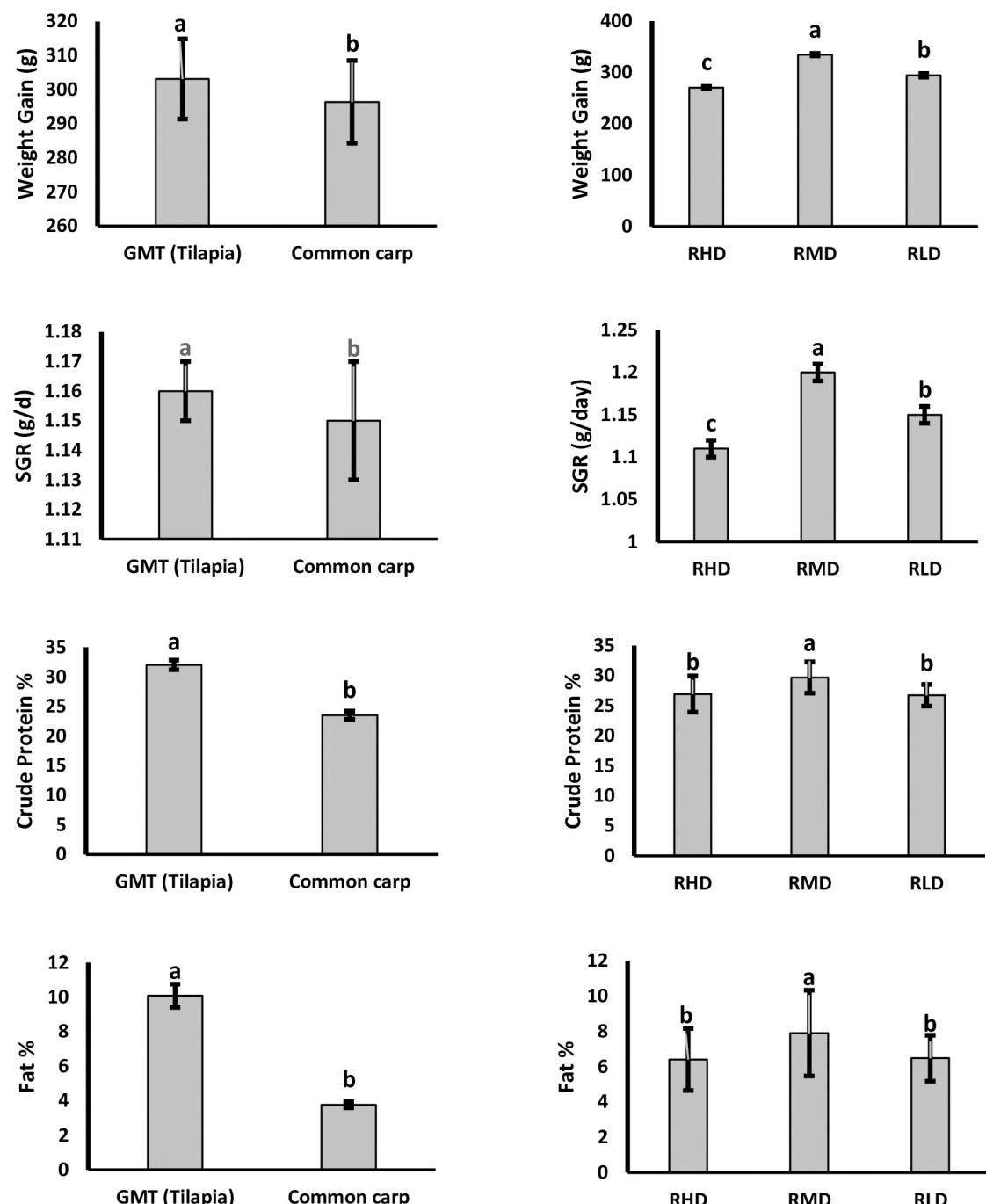

**Fig 3. Growth performance and meat proximate analysis of different treatment groups, different superscripts on bars represent significant differences at P ≤ 0.05; RHD = Rice high density (RHD), RMD = Rice medium density (RMD), RLD = Rice low density (RLD).**

RLD and significantly higher to RHD (P = 0.0196; P = 0.0166; and P = 0.0143). However, there were non-significant differences between RMD and RLD. The stem base outside diameter and panicle base outside diameter of rice were significantly higher in high-density RLD (P = 0.0110; P = 0.0035). Regarding rice stem characteristics, the medium and high rice spacing densities showed the highest values as compared to low spacing density.

**Table 1. Amino acid profile of both fish species among three different treatments.**

| AA | | GMT | | | Common carp | | | SEM | P-value | | |
|---|---|---|---|---|---|---|---|---|---|---|---|
| | | RHD | RMD | RLD | RHD | RMD | RLD | | Species | Trt | Interaction |
| EAA | Thr | 20.50[b] | 21.32[a] | 19.88[c] | 4.19[d] | 4.28[d] | 3.65[e] | 2.50 | <0.0001 | <0.0001 | <0.0001 |
| | Val | 41.85[b] | 42.35[a] | 41.87[b] | 5.01[c] | 5.08[c] | 4.42[d] | 5.61 | <0.0001 | 0.0003 | <0.0001 |
| | Met | 10.93[b] | 11.45[a] | 10.35[c] | 2.76[de] | 2.85[b] | 2.56 | 1.24 | <0.0001 | 0.0004 | <0.0001 |
| | Ile | 25.14[b] | 26.55[a] | 24.45[b] | 4.44[c] | 4.65[c] | 3.93 | 3.18 | <0.0001 | 0.0042 | <0.0001 |
| | Leu | 45.66[b] | 46.25[a] | 44.50[c] | 7.35[d] | 7.72[d] | 6.74[e] | 5.76 | <0.0001 | 0.0005 | <0.0001 |
| | Phe | 23.82[b] | 24.60[a] | 22.50[c] | 3.83[d] | 4.15[d] | 3.45[d] | 3.00 | <0.0001 | 0.0013 | <0.0001 |
| | Lys | 32.15[b] | 33.65[a] | 31.80[b] | 8.95[c] | 9.16[c] | 7.85[d] | 3.61 | <0.0001 | <0.0001 | <0.0001 |
| NEAA | Ser | 20.32[b] | 21.85[a] | 20.20[b] | 3.53[c] | 3.66[c] | 3.09[d] | 2.62 | <0.0001 | <0.0001 | <0.0001 |
| | Tyr | 24.89[b] | 25.59[a] | 24.86[b] | 2.85[c] | 3.09[c] | 2.67[c] | 3.36 | <0.0001 | 0.0256 | <0.0001 |
| | Pro | 69.73[b] | 71.51[a] | 70.14[b] | 3.12[c] | 3.04[c] | 2.61[c] | 10.18 | <0.0001 | 0.0015 | <0.0001 |
| HEAA | His | 11.88[b] | 12.85[a] | 11.35[c] | 2.90[d] | 2.92[d] | 2.75[d] | 1.39 | <0.0001 | <0.0001 | <0.0001 |
| | Arg | 132.20[ab] | 133.60[a] | 130.75[b] | 5.64[c] | 5.69[c] | 4.63[c] | 19.13 | <0.0001 | 0.0119 | <0.0001 |
| UTAA | Asp | 8.93[d] | 9.72[b] | 8.25[e] | 9.55[c] | 9.85[a] | 8.36[e] | 0.19 | <0.0001 | <0.0001 | <0.0001 |
| | Glu | 28[b].41 | 29.60[a] | 28.15[b] | 15.35[c] | 15.58[c] | 14.35[d] | 2.06 | <0.0001 | 0.0047 | <0.0001 |
| | Gly | 298.61[b] | 299.99[a] | 297.50[c] | 4.16[d] | 4.18[d] | 3.65[d] | 44.43 | <0.0001 | <0.0001 | 0.001 |
| | Ala | 174.50[b] | 181.85[a] | 179.25[a] | 5.49[c] | 5.66[c] | 4.92[c] | 26.12 | <0.0001 | <0.0001 | 0.0311 |

Means with different superscripts in row differ significantly at P ≤ 0.05; EAA = Essential Amino Acid; NEAA = Non-Essential Amino Acid; HEAA = Half Essential Amino Acid; DTAA = Umami Taste Amino Acid; RHD = Rice high density (RHD), RMD = Rice medium density (RMD), RLD = Rice low density (RLD)

## Rice yield parameter

Before the harvesting of rice, each plot of rice samples was taken to calculate the rice yield indicators (Table 3). The treatment with high density (RHD) exhibited a significantly higher yield per hectare (kg) compared to other treatment groups (P = 0.0003). Furthermore, the 1000-grain weight (G) was found to be higher in the RMD treatment, while the panicle number (ears/hill) showed higher values in the higher density (RLD) treatment, both significantly distinct from the other treatment groups (P < 0.0001; P < 0.0001). The seed setting rate, panicle setting rate, and grain no. per rice spike (Grain/ear) showed no significant differences between different treatment groups.

## Plankton community

Please refer to S1 File, which is available online, for more information on the phytoplankton and zooplankton data used in this study.

**Table 2. Rice stem characters under three treatments.**

| Traits | RHD | RMD | RLD | SEM | P-value |
|---|---|---|---|---|---|
| Plant height (cm) | 122.61[b] | 131.40[a] | 129.89[a] | 1.78 | 0.0196 |
| Stem length (cm) | 97.18[b] | 99.91[a] | 101.23[a] | 0.78 | 0.0166 |
| Panicle length (cm) | 24.23[b] | 26.79[a] | 28.46[a] | 0.80 | 0.0143 |
| Stem base outside diameter (cm) | 8.75[b] | 9.05[a] | 9.89[b] | 0.22 | 0.0110 |
| Panicle base outside diameter (cm) | 2.20[c] | 2.35[b] | 2.64[a] | 0.08 | 0.0035 |

Means with different superscripts in rows differ significantly at P ≤ 0.05; RHD = Rice high density (RHD), RMD = Rice medium density (RMD), RLD = Rice low density (RLD)

Table 3. Rice yield parameters in different treatments.

| Traits | RHD | RMD | RLD | SEM | P-value |
|---|---|---|---|---|---|
| Yield (ha/kg) | 4379.49[a] | 4265.88[b] | 4006.02[c] | 57.22 | 0.0003 |
| 1000 grain weight (g) | 20.16[c] | 20.33[b] | 20.61[a] | 0.07 | <0.0001 |
| Seed setting rate (%) | 88.09[c] | 89.15[b] | 90.48[a] | 0.36 | 0.0005 |
| Panicle setting rate (%) | 82.44[b] | 83.17[a] | 82.83[a] | 0.12 | 0.0099 |
| Grain number per rice spike (grain/year) | 339.33[a] | 311.33[c] | 327.67[b] | 4.24 | 0.0006 |
| Panicle number (year/hill) | 6.34[c] | 8.80[b] | 9.37[a] | 0.47 | <0.0001 |
| Number of hills per ha | 140251.67[a] | 104051.67[b] | 84894.67[c] | 8115.34 | <0.0001 |

Means with different superscripts in rows differ significantly at P ≤ 0.05; RHD = Rice high density (RHD), RMD = Rice medium density (RMD), RLD = Rice low density (RLD)

## Water quality parameters

Water quality parameters, including Total Dissolved Solids (TDS), Temperature, Electrical Conductivity (EC), pH, and Dissolved Oxygen (DO) were analyzed and presented in (Table 4). The data revealed that there were no significant differences between the treatments.

## Soil bio-chemical characteristics

Soil characteristics in rice-fish co-culture were examined among the treatments (Table 5). The results indicated that the organic matter content and pH value were higher in RHD and available nitrogen contents, and total nitrogen were higher in RMD whereas available phosphorus and potassium were higher in RLD however, the results were not significant among the treatments.

## Discussion

Current experimentation was tried to promote the integrated rice-cum-fish by optimizing the rice spacing density as the production of rice is related to planting of rice density. In traditional rice-growing conditions, farmers often use less space to grow rice (that is, higher human planting (20cm × 30cm) to produce high-quality rice. In an integrated rice-fish co-culture system, farmers often used a large space density of rice culture (40cm × 36cm) that favored the movement of fish but led to low rice production [33]. In the present study, the medium rice plant spacing (12×12 inch), is suggested to be best for batter rice yield with no disturbance to the fish production. Our results had shown that this medium rice spacing density could increase the efficiency of the production of rice and fish in terms of rapid growth, higher body profile, and organoleptic values.

Table 4. Water quality parameters in rice-fish co-culture among different treatments.

| Traits | RHD | RMD | RLD | SEM | P-value |
|---|---|---|---|---|---|
| Total Dissolved Solids (ppt) | 1.94 | 1.81 | 1.80 | 0.03 | 0.0756 |
| Temperature°C | 28.67 | 27.74 | 28.31 | 0.25 | 0.3743 |
| Electrical Conductivity (meter/sec) | 3.48 | 3.69 | 3.57 | 0.08 | 0.6900 |
| pH | 6.37 | 6.16 | 6.61 | 0.10 | 0.1563 |
| Dissolved Oxygen (mg/l) | 3.54 | 3.85 | 3.70 | 0.11 | 0.6584 |

RHD = Rice high density (RHD), RMD = Rice medium density (RMD), RLD = Rice low density (RLD)

**Table 5. Soil characteristics of different treatments groups.**

| Traits | RHD | RMD | RLD | SEM | *P*-value |
|---|---|---|---|---|---|
| Organic matter content (g-kg$^{-1}$) | 38.70$^a$ | 35.65$^b$ | 37.80$^{ba}$ | 0.61 | 0.0451 |
| Available Nitrogen (mg-kg$^{-1}$) | 47.45$^b$ | 53.65$^a$ | 52.30$^a$ | 1.33 | 0.0866 |
| Total Nitrogen (g-kg$^{-1}$) | 0.22$^a$ | 0.24$^a$ | 0.21$^a$ | 0.01 | 0.2258 |
| Available Phosphorus (mg-kg$^{-1}$) | 14.70$^b$ | 16.05$^{ba}$ | 17.05$^a$ | 0.48 | 0.0894 |
| Available Potassium (mg-kg$^{-1}$) | 85.05$^a$ | 86.85$^{ba}$ | 87.80$^a$ | 0.58 | 0.1015 |
| pH | 8.00$^a$ | 7.90$^a$ | 8.20$^a$ | 0.07 | 0.2448 |

Means with different superscripts in rows differ significantly at P ≤ 0.05; RHD = Rice high density (RHD), RMD = Rice medium density (RMD), RLD = Rice low density (RLD)

In rice production, fields of paddies utilize large volumes of water, which makes them suitable for aquatic farming [34]. We can find a sufficient number of aquatic animals as food and can control many problems associated with a single aquaculture system through the concurrent use of water and land resources. Optimal water temperatures play a pivotal role in enhancing the efficiency of physiological processes and biochemical reactions, aiding fish in adapting to environmental temperature variations [35]. Under the integrated system, the growth of fish was related to the DO, temperature of water, and pH. As the density of plants increases, the cover of plants rice blocks sunlight from penetrating, which leads to a lower temperature, and photosynthetic rate of plankton and consequently lowers the dissolved oxygen in water. Lower oxygen and temperature retard the growth of fish [36]. In the co-culture of fish-rice practices, traditionally farmers used a higher spacing density. This wide space will not only provide extensive space for the movements of fish but also make the environment of the paddy field more sustainable for fish growth in terms of oxygen, temperature, and water pH [31]. In the present study, different rice plant densities were tested for a rapid growth rate of fish and rice and suggested that medium density was favorable for both rice and fish with favorable factors of environment such as DO, temperature and pH that directly affect the food ingestion and indirectly affect the energy cost [37].

In our study, genetically male tilapia (GMT) showed better growth performance in medium rice density as compared to *Cyprinus carpio*. The higher weight gains and Specific Growth Rate (SGR) observed in GMT compared to *Cyprinus carpio* may be attributed to potential differences in the species' metabolic rates, genetic factors influencing growth potential, or variations in their efficiency of converting consumed feed into body mass [38]. RMD (medium rice spacing density) showed better results as compared to other treatment groups. Results indicated that rice planted in medium density had more light penetration [39] higher dissolved oxygen, high temperature, and increased diversity of phytoplankton [40], which was consumed by fish as food and directly accelerated the fish growth [41]. [42], reported that tilapia showed better growth performance LG (31.2%), SGR (42.2), and WG (5.1%) under a co-culture farming system than those of grown in ponds. [43] also describe lower values for SGR, LG, WG, and K as 9.4%, 6.7, 0.2% and 1.3%, for tilapia in the pond production system. In addition, the rich spacing had diverse microorganism flora that can be easily subjected to fish feed. Our findings support previous studies that have found that higher rice spacing densities can lead to higher yields in rice-fish culture systems. The increased space between rice plants allows for better aeration and sunlight penetration, which can lead to improved growth and productivity of both rice and fish. In addition, our results indicate that common carp and tilapia are suitable fish species for rice-fish culture systems. Studies have shown that these species have a high potential for growth and productivity in rice-fish culture systems [44].

In mixed cultures of carp and tilapia, results were better than as compared to single farming of carp. It can be specified for the efficient consumption of different niches of trophic in a field of rice [45]. [46], reported that tilapia exhibit superior growth rate and feed conversion efficiency in comparison to carp, which may indicate increased metabolic efficiency and better utilization of nutrients. This advantageous trait enables Tilapia to convert feed into body mass more effectively, highlighting its competitive edge in the utilization of available resources. This results in higher yields of fish per unit area of rice field, making tilapia more profitable. [47], reported that medium rice spacing density with lower fish stocking density was more suitable for better fish feed coefficients and improved resource utilization in a well-adjusted way. Our studies were in accordance with their studies in terms of highest growth, improved water quality parameters, food availability, and resource utilization under medium rice spacing density.

In edible organisms, proximate composition is an important factor that gives to their nutritional value in the form of protein content in muscles, ash, lipids, and moisture. These are important factors for the evaluation of muscle quality. These parameters are likely to change with environmental conditions [31]. In fish-cum-rice farming, the integration of different organisms enhances the nutrient recycling efficiency that promotes the growth of plants as well as fish and their sensorial quality. In this experiment, the crude protein and fat contents of GMT were higher as compared to *Cyprinus carpio* (P< 0.0001) and RMD (medium rice density) showed better results compared to other low and high densities. [48] Concluded that *Cyrinus carpio* showed a significant difference between crude protein and fat in different rice spacing density. High rice spacing density showed better results regarding crude protein and fat. In rice-fish co-culture, rich and diverse groups of microbes might be utilized as additional food for the growth of tilapia as compared to mono and poly-culture fish farming. The fatty acid composition of animal tissues is subject to modification in response to changes in environmental temperature. Specifically, exposure of fish to short-term temperature stress over a span of days or weeks induces alterations in the content of saturated fatty acids (SFAs), triglycerides, neutral fats, and unsaturated fatty acids (UFAs) [35]. Slight variations in the nutritional composition of fish muscles may be associated with factors related to fish feeding. These factors encompass the feeding rates of fish and the characteristics of the aquatic organisms they consume. The influence of light and temperature on both fish behavior and the availability of prey could contribute to these observed differences [49].

The muscle quality of fish and the composition of free amino acids key indicators. The content of free amino acids is the source of the increase in protein content in muscle [50]. Free amino acids are crucial for the production of protein and other nitrogen compounds involved in the regulation of synthetic and metabolic pathways [51]. In our study, the amino acid profile of GMT is much better than *Cyprinus carpio*. There was a significant difference among species comparison of various amino acids. Low rice spacing density showed the lowest value as compared to medium and high rice spacing density. In the paddy field, in the relationship between amino acid quality and quantity, the compositions of amino acids are completely fulfilling the said indicators that are producing high-level of protein. These proteins are essential for the human body due to their good taste, balanced diet, and better composition of nutrients [52]. Some of the amino acids are closely related to the quality of muscle, e.g., Ala, Gly, Pro, Arg, and Glu [48]. The flavor of muscle was also improved by amino acids and the quality of water [40]. Some of the differences between the nutritive values of the muscle of fish may be associated with the feeding of fish comprising the rates for consuming aquatic organisms under different temperatures and light intensities [53]. Our results were in accordance with the [54] as the plankton community was higher in medium rice spacing density. Like previous studies, our findings also suggested the hat quality of soil was improved due to organic matter and total nitrogen contents excreted by the fish in integrated rice-fish culture. Although, the

relationship between the nutrients, ecological/environmental factors, and rice yield is very complex, however, the improvements in yields could be increased by the proper management of line spacing density, fertilization (in terms of quality, quantity, and application mode), genetically improved seeds of rice and fish and environmental factors [55].

In Pakistan, the central Punjab region boasts a climate particularly well suited for rice cultivation, with the majority of its arable land dedicated to this crop. Rice cultivation typically commences in the optimal planting season of May to June and concludes with harvests from October to November. Research findings indicate that the maximum temperature exerts a detrimental effect on rice production, particularly during the replantation phase in June, which coincides with the vegetative growth stage. Conversely, the minimum temperature demonstrates a positive correlation with rice crops during the replantation stage. Higher minimum temperatures are associated with accelerated leaf emergence, leading to increased rice crop yields during the vegetative phase [56]. The rice spacing density is one of the most important ways to improve rice yields [57]. Based on not affecting the yield of rice, it is more likely to select a land-based farming method because it can save the cost of labor. The rice yield comprises the setting rate of seed, per panicle grain number, panicle/ unit area, and grain weight (1000). Generally, increasing the rice spacing density, which may boost the number of tillers per unit area of land, has been a deliberate and stable method of high production [22]. In our experiments, there was no significant difference among treatments. Seed setting rate, grain no. per spike and panicle setting rate are not significantly different to other treatments. Yield/ha is higher in low rice spacing density and 1000 grain weight (g) was higher in high rice spacing density.

During the growing phase, which occurs in July and August, the central Punjab's minimum temperatures have a beneficial effect on rice cultivation. This impact is particularly notable during the tillering and stem elongation stages. Furthermore, rainfall during these crucial growth phases also contributes positively to rice cultivation. It enhances the rate of tillering, ultimately leading to an increase in rice production [55]. Rice spacing density has a dual controlling effect on the growth and composition of rice]. In rice yield, the efficient utilization of nutrients in paddy fields are due to the presence of fish [58]. It was also confirmed by our study. The production of rice is higher may be due to fish excreta, high ammonia, and phosphate, utilizing the N directly -[48] and too easily taken up with rice in the form of potassium. The present study showed that, in rice-fish co-culture fishes play an important role in increasing the production and yield of rice on the potential impact of fertilization of fish excreta. Implementing a well-balanced environmental plan in rice-fish culture highlights the sustainable role of fish that adapt to local conditions.

Optimization of scientific experiments can help to increase the interest of local farmers in promoting and enriching the fish rice tradition. Our experiment showed that medium rice density could be converted into feed-based value, growth rates, muscle quality, soil biochemistry, plankton community, rice stem characteristics, and rice yield parameters. Numerous studies are needed to expose certain ways of distinguishing fish amino acids and fatty acids from the various congestion of rice cultivation. The composition of a healthy fish diet is associated with changes in body length and weight and further research, in addition, will be needed to illustrate this issue. If we can apply the philosophy behind these traditional systems, future agriculture may have more opportunities to meet additional food needs while protecting the environment time. Based on the scientific research behind them, we can take better advantage of this farming system while creating more value, they can be conserved dynamically. Consequently, the implementation of a fish-rice culture can increase the production and economics of farmers [59]. As a result, sustainable farming can be achieved by all natural resources in rice-fish co-culture, which can remove poverty and increase the economics of farmers.

## Conclusion

It has been concluded that the performance of both rice and fish is improved by sustainable optimized management approaches. Co-culture of both carp and tilapia in different rice spacing densities showed higher fish and rice production, and improved meat/rice quality. Additionally, this approach effectively balances soil characteristics, primary production, and resource utilization demonstrating its potential for sustainable agriculture. The results indicated that medium rice spacing density in the rice-fish co-culture system was the most effective eco-culture model for the promotion of both fish and rice. As the co-culture system is adopted for efficient resource utilization, the government must urge farmers to use technology by providing farmers with the necessary technical knowledge and additional benefits to improve the lives of rural farmers. It is recommended that a high yield of rice and quality of GMT in the participatory system could be an attraction for consumers and support the sustainable development of GMT co-culture. However, to ensure the long-term success and sustainability of this co-culture, it is imperative to incorporate effectively parasite management strategies into an overall management plan. These strategies may include regular monitoring, appropriate treatment usage, and crop rotation. The optimum fertilizer/feed, stocking density of fish species, economic optimum in different rice spacing densities and emphasizing the importance of parasite management will be investigated in further studies. This would help in Pakistan as well as other rice-producing countries to keep up with the growing demand for fish and generate income in rural areas.

## Supporting information

**S1 File.**
(DOCX)

## Acknowledgments

The authors extend their appreciation to the Syed Ghulam Mohayudin, Lecturer, and Incharge Remote Sensing and Geographic Information System (GIS) Laboratory, Department of Wildlife & Ecology, University of Veterinary and Animal Sciences, Lahore, Pattoki, Ravi Campus, Punjab, Pakistan for help in designing the study area map.

## Author Contributions

**Conceptualization:** Farzana Abbas.

**Data curation:** Muhammad Inayat.

**Formal analysis:** Muhammad Inayat.

**Investigation:** Muhammad Hafeez-ur-Rehman.

**Project administration:** Muhammad Hafeez-ur-Rehman.

**Resources:** Muhammad Hafeez-ur-Rehman.

**Software:** Athar Mahmud.

**Supervision:** Farzana Abbas.

**Writing – original draft:** Muhammad Inayat.

**Writing – review & editing:** Athar Mahmud.

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
