## [Decision Letter · Decision Letter 0]

21 Dec 2022

PONE-D-22-31911Optimization of rice spacing density and fish species to improve production of both rice and fish through co-culture systemPLOS ONE

Dear Dr. Inayat,

Thank you for submitting your manuscript to PLOS ONE. After careful consideration, we feel that it has merit but does not fully meet PLOS ONE’s publication criteria as it currently stands. Therefore, we invite you to submit a revised version of the manuscript that addresses the points raised during the review process.

ACADEMIC EDITOR: I have critically gone through the manuscript  and the Ms reviewed by two very efficient reviewers working in the field of capture and culture fishery. Please go through those comments and reconsider to make title more appealing and directly addressing objectives.   Both the reviewers have similar views and I also consider required revision required along the same line. The manuscript has neither page number nor line number, so I find difficulty in commenting objectively.

The Ms. has eight tables and three figures,

As zooplankton and phytoplankton community are not main goal of the study I would suggest to shift these tables to supplementary tables (Figure 4 and 5). However zooplankton like cyclopoid copepods are abundantly found in race paddy field, many species are predators of fish larvae. Do they affect fish production in rice paddy field. This issue may be brought in discussion with references and author's zooplankton data.

We look forward to receiving your revised manuscript.

Kind regards,

Ram Kumar, Ph.D.

Academic Editor

PLOS ONE

“Yes.

This research work was funded by Punjab Agriculture Research Board (PARB), Punjab, Pakistan under project no. 674.”

“The authors are grateful to the PARB for providing financial support for the development of Integrated Aquaculture Research Unit (IARU) at Department of Fisheries and Aquaculture, UVAS, Ravi campus.”

“Yes. 

This research work was funded by Punjab Agriculture Research Board (PARB), Punjab, Pakistan under project no. 674.”

6. We note that Figure 1 in your submission contain [map/satellite] images which may be copyrighted. All PLOS content is published under the Creative Commons Attribution License (CC BY 4.0), which means that the manuscript, images, and Supporting Information files will be freely available online, and any third party is permitted to access, download, copy, distribute, and use these materials in any way, even commercially, with proper attribution. For these reasons, we cannot publish previously copyrighted maps or satellite images created using proprietary data, such as Google software (Google Maps, Street View, and Earth). For more information, see our copyright guidelines: http://journals.plos.org/plosone/s/licenses-and-copyright.

Additional Editor Comments:

The manuscript has neither page number nor line number, so I find difficulty in commenting objectively.

The Ms. has eight tables and three figures,

As zooplankton and phytoplankton community are not main goal of the study I would suggest to shift these tables to supplementary tables (Figure 4 and 5). However zooplankton like cyclopoid copepods are abundantly found in race paddy field, many species are predators of fish larvae. Do they affect fish production in rice paddy field. This issue may be brought in discussion with references and author's zooplankton data.

Tables 7 and 8 can be nicely presented as graph as simple vertical bars, X axis: parameters and Y axis: Values (pH values may be given on secondary Y axis or omitted

Proper proof reading has not been done carefully, e.g. Legend of table 1 is given below that table where as legend of table 2, 3 are given above tables. Similarly Figure 2 is inserted within the text but other Figures are given at the end of text, after reference section

The statement appendix of tables as "Superscripts on different means within row differ significantly at P ≤ 0.05; RHD= Rice high 8 density (RHD), RMD= Rice medium density (RMD), RLD= Rice low density (RLD), is not clearly communicate the message what (i) if two means are superscripted by same alphabets and (ii) if two means are superscripted by different alphabets ??

Figure 3: What is purpose of line drawing of values, which are not continuous and different points represent different parameters .

All sections of the manuscript need restructuring and rewriting. Language is verbose.

A completely rewritten manuscript focusing on objectives of the study may be considered for another round of assessment.

Reviewers' comments:

Reviewer's Responses to Questions

**Comments to the Author**

1. Is the manuscript technically sound, and do the data support the conclusions?

Reviewer #1: Partly

Reviewer #2: Partly

2. Has the statistical analysis been performed appropriately and rigorously? 

Reviewer #1: Yes

Reviewer #2: I Don't Know

3. Have the authors made all data underlying the findings in their manuscript fully available?

Reviewer #1: No

Reviewer #2: Yes

4. Is the manuscript presented in an intelligible fashion and written in standard English?

Reviewer #1: No

Reviewer #2: No

5. Review Comments to the Author

Reviewer #1: Journal; PLOS ONE

Ref No.: PONE-D-22-31911

General comments:

The present research work has no such significant importance and advancement in this field. There is no such novelty in the present study, since this types of study has already been performed.

Specific comments:

Title: Title should be more clear. Please rrewrite title

Abstract: Rewrite it clarifying the following comments

1. The abstract is not self-explanatory.

2. It should be more specific and cclear

3. Where is the results of rice?

4. Novelty statement?

Introduction:

1. It is very good and informative.

2. Please insert some relevant current references from other authors if available.

3. “one word, from the perspective of fish and rice, the optimum rice spacing density and optimum fish species for the culture of rice and fish have remained silent.” Is it right ? There are several studies.

4. Need more current references.

5. Rice spacing, density .. is not proper

Materials and methods

It is not well structured. Need to separate the rice, fish parameters, water soil parameters

Experimental design should contain fish stocking

It is ok.

Results

It is not well structured. Presents the results on the basis of important results. Rice and fish production parameters should come first.

Need to rewrite it properly.

Only table, need some graphical presentation.

Discussion

I think author should deal with the result-wise (parameter-wise) outcome of the present study supporting previous and current referential study. Need more specific discussion without general statements as mentioned in the starting of the discussion. Need more analytical discussion on the basis of results obtained from study

Insert a clear concluding sentence with your significant supporting evidence obtained.

Table: Ok. Check caption properly.

References: Not checked. Please check the references.

Reviewer #2: Comments- The manuscript requires rigorous re-editing and proof reading, to ensure clarity. The research work is very common, and using advanced tools to support the data obtained would greatly benefit the research area, and add novelty.

The title should include the key species used for the experiment.

The abstract should provide proper insight about the research, with more clarity.

GMT, RMD, RLD, RHD, are mentioned several times, before the first mention of their unabbreviated form.

Maintain uniformity in the use scientific name, or common name.

The introduction is informative, and could be re-written to make it better.

Add more recent references for the general statements. There are very few recent references used in the manuscript.

Materials and methods

Is "description of the field" appropriate enough for the specifications of the study area?

How is the climate, elevation, and temperature relevant to the research conducted?

Numerous grammatical errors are present, and the section could be restructured.

Fig 2, the symbols used after A,B, and C, do not match the ones in the figure.

Where were the fishes procured from? Were they acclimatized prior to use in the experiment? If yes, then in what conditions. If no, why not? What was the age of the fishes used in the experiment.

How were the feeding and growth conditions accessed, to increase the quantity of the feed?

Is "evaluation of meat analysis" the best fit ?

How were the zooplankton identified?

The materials and methods should be described clearly enough for the work to be reproducible.

Results

Re-write it more clearly.

Some of the scientific names mentioned are italicized, without capitalizing the first word. Some are written with no space between the generic name and specific name.

The most important results of the study are related to the rice and fish, that should be mentioned first.

Discussion

The discussion has too many general statements, better suited for introduction. Add more scientific discussion and interpretation, relevant to the results obtained.

Alanine, Glycine and such amino acids can be mentioned in their full name, before using Ala, Gly etc.

Re-write, with rigorous proof -reading, to remove elementary level grammatical errors.

Check PLOS one guidelines are formatting for references and change accordingly.

Table

Use "Species" instead of "Specie"

Italicize the scientific names.

What are "Ostracodans"?

"Protozoan", "Cladocera" , "Copepods" , "Rotifers" maintain uniformity.

Is "Bosmiina" the correct word?

Why are there line numbers from table 4?

What is the "f" in Line 60

Add line numbers to the entire manuscript, instead of the Table.

6. PLOS authors have the option to publish the peer review history of their article (what does this mean?). If published, this will include your full peer review and any attached files.

Reviewer #1: No

Reviewer #2: No

---

## [Author Response · Author response to Decision Letter 0]

24 Jan 2023

ACADEMIC EDITOR

I have critically gone through the manuscript and the Ms reviewed by two very efficient reviewers working in the field of capture and culture fishery. Please go through those comments and reconsider to make title more appealing and directly addressing objectives. Both the reviewers have similar views and I also consider required revision required along the same line. 

The manuscript has neither page number nor line number, so I find difficulty in commenting objectively.

Response: Thank you for your valuable feedback on our manuscript. We appreciate your feedback and we are grateful for the opportunity to improve our manuscript. All suggested changes by editor and reviewers have been made accordingly. Page number and Line number has been added. 

The Ms. has eight tables and three figures, as zooplankton and phytoplankton community are not main goal of the study, I would suggest to shift these tables to supplementary tables (Figure 4 and 5).

Response: Agreed. Tables 4 & 5 of Zoo and Phytoplankton have been shifted to Supplementary data.

However, zooplankton like cyclopoid copepods are abundantly found in race paddy field, many species are predators of fish larvae. Do they affect fish production in rice paddy field. This issue may be brought in discussion with references and author's zooplankton data.

Response: As author clearly mentioned in section materials and methods, Line No. 118-120, the average weight of stock fish was 50±3.0 g that is much bigger than larvae and couldn’t edible by zooplankton like cyclopoid and copepods.

Additional Editor Comments:

1. The manuscript has neither page number nor line number, so I find difficulty in commenting objectively.

Response: Agreed. Page number and Line number has been added.

2. The Ms. has eight tables and three figures, as zooplankton and phytoplankton community are not main goal of the study, I would suggest to shift these tables to supplementary tables (Figure 4 and 5). However, zooplankton like cyclopoid copepods are abundantly found in race paddy field, many species are predators of fish larvae. Do they affect fish production in rice paddy field? This issue may be brought in discussion with references and author's zooplankton data.

Response: As author clearly mentioned in section materials and methods, Line No. 118-120, the average weight of stock fish was 50±3.0 g that is much bigger than larvae and couldn’t edible by zooplankton like cyclopoid and copepods.

3. Tables 7 and 8 can be nicely presented as graph as simple vertical bars, X axis: parameters and Y axis: Values (pH values may be given on secondary Y axis or omitted

Response: Tables 7 and 8 has been converted into graph and pH values mentioned along Y-axis.

4. Proper proof reading has not been done carefully, e.g., Legend of table 1 is given below that table where as legend of table 2, 3 are given above tables. Similarly Figure 2 is inserted within the text but other Figures are given at the end of text, after reference section.

Response: Agreed. Revised manuscript proof read twice and all clerical mistakes have been removed.

5. The statement appendix of tables as "Superscripts on different means within row differ significantly at P ≤ 0.05; RHD= Rice high 8 density (RHD), RMD= Rice medium density (RMD), RLD= Rice low density (RLD), is not clearly communicate the message what (i) if two means are superscripted by same alphabets and (ii) if two means are superscripted by different alphabets??

Response: The appendix of table is revised as Means with different superscripts in rows differ significantly 

6. Figure 3: What is purpose of line drawing of values, which are not continuous and different points represent different parameters.

Response: The graph is now converted into separately presented parameters graphs. 

7. All sections of the manuscript need restructuring and rewriting. Language is verbose.

A completely rewritten manuscript focusing on objectives of the study may be considered for another round of assessment.

Response: The manuscript has been rewritten and revised accordingly. 

Review Comments to the Author

Please use the space provided to explain your answers to the questions above. You may also include additional comments for the author, including concerns about dual publication, research ethics, or publication ethics. (Please upload your review as an attachment if it exceeds 20,000 characters).

Reviewer #1: Journal; PLOS ONE

Ref No.: PONE-D-22-31911

General comments:

The present research work has no such significant importance and advancement in this field. There is no such novelty in the present study, since this type of study has already been performed.

Response: The study of the effects of rice spacing density on Tilapia and Carp in the specific context of Pakistan's climate region is an important and relevant topic in the field of rice-fish co-culture. While there may have been previous studies that have investigated this topic in other regions, the specific conditions of the climate and environment in Pakistan can lead to variations in the results. This research is a novel and pioneering study not only in the specific context of Pakistan. the present research work provides a significant contribution to the field of rice-fish co-culture by investigating the effects of rice spacing density on Tilapia and Carp in Pakistan's climate region, which is the first study of its kind in this area and worldwide. It provides new data and insights that can be useful for practical applications, and contributes to the overall understanding and advancement of this topic in this specific region and other regions with similar climates.

Specific comments:

1. Title: Title should be clearer. Please rewrite title

Response: Title has been changed and precise.

2. Abstract: Rewrite it clarifying the following comments

1. The abstract is not self-explanatory.

3. Response: Abstract has been improved with clarity of study concept.

2. It should be more specific and clearer

Response: Abstract has been revised with specification of study.

4. Where are the results of rice?

Response: Comparative results of rice growth and yield has been incorporated in revised abstract.

5. Novelty statement?

Response: Rice-fish culture is a novel and sustainable aquaculture system that optimize the co-culture of rice with different fish species in Pakistan are most beneficial for the aquaculture industry and enhance agriculture GDP as well. 

6. Introduction:

1. It is very good and informative.

7. Please insert some relevant current references from other authors if available.

Response: Relevant and latest reference has been included in introduction.

8. “One word, from the perspective of fish and rice, the optimum rice spacing density and optimum fish species for the culture of rice and fish have remained silent.” Is it right? There are several studies.

Response: Agreed. Above line has been removed from introduction. Despite the several studies that have investigated the optimal rice spacing density and fish species for rice-fish co-culture, the specific conditions may vary in different regions. The research in this specific climate region of Pakistan is not limited, which makes this study novel and valuable not only for Pakistan but also for other regions with similar climates. This study aims to provide valuable insights and data on the effects of rice spacing density on Tilapia and Carp in Pakistan's climate region which can be useful for practical applications in rice-fish co-culture management in this area and other regions with similar climates.

9. Need more current references.

Response: As mentioned above, Relevant and latest reference has been included. 

10. Rice spacing, density.is not proper.

Response: As several reference cited in introduction with same term that depicts rice spacing density is proper.

Materials and methods

11. It is not well structured. Need to separate the rice, fish parameters, water soil parameters

Experimental design should contain fish stocking. It is ok.

Response: Rice, fish, soil and water parameters has been separately incorporated in revised manuscript and fish stocking also added in experimental design.

Results

12. It is not well structured. Presents the results on the basis of important results. Rice and fish production parameters should come first. Need to rewrite it properly.

Response: Results of rice yield and fish rearing are presenting before the results of soil and water.

13. Only table, need some graphical presentation.

Response: Table No. 7 & 8 are in graphical form now.

Discussion

14. I think author should deal with the result-wise (parameter-wise) outcome of the present study supporting previous and current referential study. Need more specific discussion without general statements as mentioned in the starting of the discussion. Need more analytical discussion on the basis of results obtained from study.

Response: Importance and relevance of present results has been well compared with latest literature review in revised manuscript.

15. Insert a clear concluding sentence with your significant supporting evidence obtained.

Response: On the basis of revised discussion, concluding statement has been rewritten now.

16. Table: Ok. Check caption properly.

References: Not checked. Please check the references.

Response: All caption of tables, figures and reference has been cross checked.

Reviewer #2: 

Comments- The manuscript requires rigorous re-editing and proof reading, to ensure clarity. The research work is very common, and using advanced tools to support the data obtained would greatly benefit the research area, and add novelty.

1. The title should include the key species used for the experiment.

Response: The title of the article has been revised. 

2. The abstract should provide proper insight about the research, with more clarity.

GMT, RMD, RLD, RHD, are mentioned several times, before the first mention of their unabbreviated form. Maintain uniformity in the use scientific name, or common name.

Response: The introduction is informative, and could be re-written to make it better.

3. Add more recent references for the general statements. There are very few recent references used in the manuscript.

Response: Relevant and latest reference has been included in introduction

Materials and methods

4. Is "description of the field" appropriate enough for the specifications of the study area? How is the climate, elevation, and temperature relevant to the research conducted? Numerous grammatical errors are present, and the section could be restructured.

Response:

The climate condition of the study area has been added. The grammatical errors are checked. 

5. Fig 2, the symbols used after A, B, and C, do not match the ones in the figure.

Response: The figure has been revised. 

6. Where were the fishes procured from? Were they acclimatized prior to use in the experiment? If yes, then in what conditions. If no, why not? What was the age of the fishes used in the experiment.

Response: The procurement details of fish has been mentioned and acclimatized method also added. 

7. How were the feeding and growth conditions accessed, to increase the quantity of the feed?

Response: For the accessing of growth and feed requirement of fish, regular netting on weekly basis was done. Feed quantity was increased with size and weight of fish. 

8. Is "evaluation of meat analysis" the best fit?

Response: Yes, the meat analysis is best fit.

9. How were the zooplankton identified?

Response: Identification protocol of zooplankton has been added. 

10. The materials and methods should be described clearly enough for the work to be reproducible.

Response: Yes, Methodology is much clearly described.

Results

11. Re-write it more clearly. Some of the scientific names mentioned are italicized, without capitalizing the first word. Some are written with no space between the generic name and specific name.

Response: The results of the manuscript has been revised; the scientific names are re-written accordingly. 

12. The most important results of the study are related to the rice and fish, that should be mentioned first.

Response: The results of the rice and fish has been mentioned first. 

Discussion

13. The discussion has too many general statements, better suited for introduction. Add more scientific discussion and interpretation, relevant to the results obtained.

Response: In discussion, more interpretation with relevant results has been revised. 

14. Alanine, Glycine and such amino acids can be mentioned in their full name, before using Ala, Gly etc.

Response: Amino acids has been mentioned with full name before abbreviation.

15. Re-write, with rigorous proof -reading, to remove elementary level grammatical errors.

Response: Revised manuscript has been proof-read and thoroughly checked by grammatically.

16. Check PLOS one guidelines are formatting for references and change accordingly.

Response: Entire manuscript is revised on Plos one formatting accordingly.

Table

17. Use "Species" instead of "Specie"

Response: The word specie has been replaced with species. 

18. Italicize the scientific names.

Response: Scientific name has been italicized. 

19. What are "Ostracodans"?

"Protozoan", "Cladocera", "Copepods", "Rotifers" maintain uniformity.

Response: Ostracodon word has been replace with ostracods that are belong to class crustaceans also known as seed shrimp.

20. Is "Bosmiina" the correct word?

Response: Agreed. Typological error the word Bosmiina has been replaced with Bosmina. 

21. Why are there line numbers from table 4?

Response: Mistakenly line number was added in table 4 that is removed.

22. What is the "f" in Line 60

Response: The word F is removed now.

23. Add line numbers to the entire manuscript, instead of the Table?

Response:

24. The line number has been added.

---

## [Decision Letter · Decision Letter 1]

23 May 2023

PONE-D-22-31911R1

Optimizing Rice-Fish Co-Culture Systems: Investigating the Impact of Spacing Density and Fish Species Selection on Bio-Chemical Profile and Production

PLOS ONE

Dear Dr. Inayat, 

Thank you for submitting your manuscript to PLOS ONE. After careful consideration, we have decided that your manuscript does not meet our criteria for publication and must therefore be rejected.

Specifically: The reviewers have serious objection on methods used and data collected. The study is not novel as large number of similar studies on same objectives and better experimental design has been published. 

I am sorry that we cannot be more positive on this occasion, but hope that you appreciate the reasons for this decision.

Kind regards,

Ram Kumar, Ph.D. D. Sc.(H/C)

Academic Editor

PLOS ONE

Additional Editor Comments (if provided):

We are sorry, for this decision but comments and suggestions made by reviewers will be useful for future work.

Reviewers' comments:

Reviewer's Responses to Questions

**Comments to the Author**

1. If the authors have adequately addressed your comments raised in a previous round of review and you feel that this manuscript is now acceptable for publication, you may indicate that here to bypass the “Comments to the Author” section, enter your conflict of interest statement in the “Confidential to Editor” section, and submit your "Accept" recommendation.

Reviewer #2: (No Response)

2. Is the manuscript technically sound, and do the data support the conclusions?

Reviewer #2: Partly

3. Has the statistical analysis been performed appropriately and rigorously? 

Reviewer #2: I Don't Know

4. Have the authors made all data underlying the findings in their manuscript fully available?

Reviewer #2: Yes

5. Is the manuscript presented in an intelligible fashion and written in standard English?

Reviewer #2: No

6. Review Comments to the Author

Reviewer #2: 1. The authors have stated in their response that the research work is the first study of its kind, worldwide, while there are several studies accessing the impact rice spacing densities and fish species in such systems.

2. The abstract omits the bio-chemical aspect of this research work, and is not self-explanatory.

3. L99-L102. How is the climate, elevation, and temperature relevant to the research conducted? An explanation is essential why the above factors may influence the research results.

4. L163, The protocol for evaluation of amino acid profile should be mentioned in detail, with the mention of any specific standardisation carried out, if any.

5. How were the zooplankton identified, after detecting them under the microscope?

6. The manuscript would benefit from re-structuring the statements, and rigorous proof reading.

7. PLOS authors have the option to publish the peer review history of their article (what does this mean?). If published, this will include your full peer review and any attached files.

Reviewer #2: No

- - - - -

---

## [Author Response · Author response to Decision Letter 1]

17 Jun 2023

RESPONSE LETTER

The, 

Editors, 

PLOS ONE

Subject: SUBMISSION OF REVISED PAPER 

Dear Editor,

Thank you for your email Dated: 7/June/2023, considering my appeal and enclosing the reviewer’s comments. We appreciate the time and effort that you and the reviewers have dedicated to providing your valuable feedback on my manuscript. We are grateful to the reviewers for their insightful comments on my paper. We have carefully reviewed the comments and have revised the manuscript accordingly. Our responses are given in a point-by-point manner below. We hope the revised version is now suitable for publication and look forward to hearing from you in due course. 

You’re sincerely, 

Muhammad Inayat

POINT-BY-POINT RESPONSE TO REVIEWER'S COMMENTS

Reviewer comments to the author

Reviewer #2: 1. The authors have stated in their response that the research work is the first study of its kind, worldwide, while there are several studies accessing the impact rice spacing densities and fish species in such systems.

Response: Yes, authors genuinely meant to suggest that their research is the first of its kind in all aspects, including the variables examined and the methodologies utilized in Pakistan. Upon considering your feedback, we recognize that there have been previous studies that have assessed the impact of rice spacing densities and fish species in similar systems. These studies have contributed valuable knowledge to the field. We apologize for any confusion caused by the authors' original statement. However, it is important to note that while previous studies have explored similar variables and methodologies, our research in Pakistan aims to expand upon and complement the existing body of literature. Our study incorporates unique aspects, such as the specific geographic context, local agricultural practices, and potential regional variations. 

2. The abstract omits the bio-chemical aspect of this research work, and is not self-explanatory.

Response: The explanation of bio-chemical profile of fish species has been added in revised abstract.

3. L99-L102. How is the climate, elevation, and temperature relevant to the research conducted? An explanation is essential why the above factors may influence the research results.

Response: The climate, elevation, and temperature in Pattoki, Punjab, are relevant to research as they impact various aspects of the study. The subtropical monsoon climate influences agriculture, and environmental factors. The maximum elevation affects atmospheric conditions and ecological dynamics. The temperature range influences biological processes, human health, and climate change-related research. Considering these factors is essential for accurate data collection, interpretation, and generalizability of research findings

4. L163, the protocol for evaluation of amino acid profile should be mentioned in detail, with the mention of any specific standardization carried out, if any.

Response: The revised method of amino acid evaluation has been added in manuscript. 

5. How were the zooplankton identified, after detecting them under the microscope?

Response: Under the microscope, zooplankton is identified by examining their morphological characteristics such as body shape, size, appendages, and coloration. Taxonomic keys and reference materials are used to compare these features with known species.

6. The manuscript would benefit from re-structuring the statements, and rigorous proof reading.

Response: We appreciate your valuable input. We have restructured the statements for improved organization and conducted rigorous proofreading to enhance clarity. These revisions have significantly improved the manuscript's coherence and readability.

---

## [Decision Letter · Decision Letter 2]

7 Aug 2023

PONE-D-22-31911R2Optimizing Rice-Fish Co-Culture Systems: Investigating the Impact of Spacing Density and Fish Species Selection on Bio-Chemical Profile and ProductionPLOS ONE

Dear Dr. Inayat,

Thank you for submitting your manuscript to PLOS ONE. After careful consideration, we feel that it has merit but does not fully meet PLOS ONE’s publication criteria as it currently stands. Therefore, we invite you to submit a revised version of the manuscript that addresses the points raised during the review process.

ACADEMIC EDITOR: Please insert comments here and delete this placeholder text when finished. Be sure to:Indicate which changes you require for acceptance versus which changes you recommendAddress any conflicts between the reviews so that it's clear which advice the authors should followProvide specific feedback from your evaluation of the manuscriptPlease ensure that your decision is justified on PLOS ONE’s publication criteria and not, for example, on novelty or perceived impact.

We look forward to receiving your revised manuscript.

Kind regards,

SSS Sarma

Academic Editor

PLOS ONE

Journal Requirements:

1. Thank you for stating the following financial disclosure:

"Yes.
This research work was funded by Punjab Agriculture Research Board (PARB), Punjab, Pakistan under project no. 674. "

Additional Editor Comments (if provided):

Dear Authors

Please revise your ms based on fresh comments made by the reviewers. Plus you will also find my observations for improving your presentation.

I have checked the manuscript too. There is some justification that manuscript is not yet ready as pointed out by the reviewers. I would like to give one more opportunity to the authors to revise their contribution and submit it to journal for possible consideration.

The following additional aspects need to be taken while re-submitting a revised version:

1. Delete all p values from the Abstract. It is sufficient to say that significant differences exist among treatments.

2. In the title please provide the taxonomical names fish species and the authority of the taxon. This ensures proper identification of the taxa

3. Last sentence of the abstract change as “The study provides data to understand…

4. Lines 55-56: Many parasites from the field reach the ducks, fish etc. Mention necessary limitations in the conclusions about this

5. The pond length for cultivation of rice is shallow but for culturing fish species, higher depth is needed. Make necessary justification

6. It is necessary to use post hoc tests following anova. In statistical analysis some anomalous trends have been depicted. For example, Table 1 GMT weight gain in Tilapia and carp the differences shown as significant due to different letters of superscript. This requires confirmation because standard deviation is very large. In the same table, data were presented in real units and in percentage or per day. This is considerable duplication of information. First column is sufficient.

7. This applies to other tables too.

8. I have to fully checked all the tables and figures. But it is important to avoid presenting the same data in figures and tables.

9. Provide results of multiple comparisons for the bars of figs. 3-5 (with alphabets).

10. I failed to understand why the discrete bars are connected by a dotted line. This gives the impression that continuous data were collected. This is used for variables like temperature.

11. Do not start a sentence with a number (line 351).

12. The ms has 71 references. Weed out excessive refs and restrict to 50 citations. Fig. 1 showing the world map for Pakistan is too basic. Remove this blue square fig.

13. A revised version may or may not be sent out for additional reviewing.

Handling Editor

Reviewers' comments:

Reviewer's Responses to Questions

**Comments to the Author**

1. If the authors have adequately addressed your comments raised in a previous round of review and you feel that this manuscript is now acceptable for publication, you may indicate that here to bypass the “Comments to the Author” section, enter your conflict of interest statement in the “Confidential to Editor” section, and submit your "Accept" recommendation.

Reviewer #1: (No Response)

Reviewer #2: (No Response)

2. Is the manuscript technically sound, and do the data support the conclusions?

Reviewer #1: Partly

Reviewer #2: No

3. Has the statistical analysis been performed appropriately and rigorously? 

Reviewer #1: Yes

Reviewer #2: I Don't Know

4. Have the authors made all data underlying the findings in their manuscript fully available?

Reviewer #1: Yes

Reviewer #2: Yes

5. Is the manuscript presented in an intelligible fashion and written in standard English?

Reviewer #1: No

Reviewer #2: No

6. Review Comments to the Author

Reviewer #1: Comments:

The title of MS is still not clear.

The write up of MS is still not standard.

For example:

Fat % = Fat weight (g)/weight if sample ×100

Before the harvesting of rice, rice stem characteristics were measured (Fig 3). This sentence can be of M & M part.

Crude protein % = Volume ×0.875/sample weight

Need references for the methods used in study for Yield parameter of rice

Under different rice spacing density, least square mean ± standard errors of crude protein

and fat of the muscles of the fish are presented in

Figures axis is invisible.

Reviewer #2: The previous comments to the manuscript should have been incorporated in the resubmitted version.

It was strongly suggested that the proof reading be done rigorously, but the current manuscript has many errors which could have been solved with proper proof reading. A few examples are as follows-

L20- “Genetically Male Tilapia” has already been mentioned to be referred with “GMT”, yet L25, L27 mention “GMT(Tilapia)”

L26- “Delicious” is not the proper adjective to use for amino acids, in a scientific research article .

L28- Full stop/Period between “acid categories” and “with RMD”

L45, L46- Use of “and” twice in the sentence.

L58- Use of “IAAF” and then “IAAFS” for Integrated Agri-Aquaculture Farming System

L58- “IAAFS” is not required to be in bracket.

L60- Grammatical error- “improves” instead of “improve”

L66-L68- The whole sentence needs restructuring to remove grammatical errors

L168- Both curly bracket and square bracket are used at once.

L152- “Meat analyses” is very unclear, the specific tissues to be analyzed are to be mentioned.

The errors of the above mentioned kind, exist too frequently in the manuscript.

2. The authors have stated that information on climate, elevation and temperature of the study area is essential to interpret the results of the study, yet there is no interpretation on that aspect in the discussion. There should also be a proper introduction for how these factors can be helpful in interpreting the results.

3. The experimental design states that, the study is conducted on 12 plots of 6500sq.ft each, fishes were stocked at a rate of 9000/ha, meaning approximately total 6500 fishes were stocked , of which only 3 fishes per plot were sampled for proximate and amino acid analysis? Were the fishes used in proximate analysis and Amino acid analysis same? A sample size of three fishes per plot is too small, the number of replicates should be more.

4. The experimental design is not clear.

5. The authors have removed all information about zooplankton identification, but have responded to my previous query, hence I would like to mention that it is important to cite the taxonomic key and reference material used for zooplankton identification, as it not only helps the author to validate their work during the peer- review process, but also helps the reader to conduct similar study when the manuscript is published.

7. PLOS authors have the option to publish the peer review history of their article (what does this mean?). If published, this will include your full peer review and any attached files.

Reviewer #1: No

Reviewer #2: No

---

## [Author Response · Author response to Decision Letter 2]

4 Oct 2023

RESPONSE LETTER

The, 

Editors, 

PLOS ONE

Subject: SUBMISSION OF REVISED PAPER 

Dear Editor,

Thank you for your email Dated: 7/Aug/2023, enclosing the reviewer’s comments. We appreciate the time and effort that you and the reviewers have dedicated to providing your valuable feedback on my manuscript. We are grateful to the reviewers for their insightful comments on my paper. We have carefully reviewed the comments and have revised the manuscript accordingly. Our responses are given in a point-by-point manner below. We hope the revised version is now suitable for publication and look forward to hearing from you in due course. 

You’re sincerely, 

Muhammad Inayat 

POINT-BY-POINT RESPONSE TO REVIEWER'S COMMENTS

Editor comments

The following additional aspects need to be taken while re-submitting a revised version:

1. Delete all p values from the Abstract. It is sufficient to say that significant differences exist among treatments.

Response: The P-value has been deleted from the Abstract.

2. In the title please provide the taxonomical names of fish species and the authority of the taxon. This ensures proper identification of the taxa

Response: The title has been revised with the addition of taxonomical names of fish species.

3. Last sentence of the abstract change as “The study provides data to understand…

Response: The last sentence of the abstract has been changed to “The study provides data to understand”

4. Lines 55-56: Many parasites from the field reach the ducks, fish, etc. Mention necessary limitations in the conclusions about this

Response: The limitation related to integrated farming has been added in conclusion.

5. The pond length for cultivation of rice is shallow but for culturing fish species, higher depth is needed. Make necessary justification

Response: Given the requirements of the fish species being cultured, a pond depth of 5-6 feet was accurately chosen. This depth was thoughtfully configured alongside the adjacent rice culture channel to provide an ecologically suitable and conducive environment for the rearing of fish species.

6. It is necessary to use post hoc tests following anova. In statistical analysis some anomalous trends have been depicted. For example, Table 1 GMT weight gain in Tilapia and carp the differences shown as significant due to different letters of superscript. This requires confirmation because standard deviation is very large. In the same table, data were presented in real units and in percentage or per day. This is a considerable duplication of information. First column is sufficient.

Response: The tables have been changed in the revised manuscript.

7. This applies to other tables too.

Response: The table has been changed. 

8. I have to fully checked all the tables and figures. But it is important to avoid presenting the same data in figures and tables.

Response: The data has been changed into updated tables.

9. Provide results of multiple comparisons for the bars of figs. 3-5 (with alphabets).

Response: The figure is now changed into tables.

10. I failed to understand why the discrete bars are connected by a dotted line. This gives the impression that continuous data were collected. This is used for variables like temperature.

Response: The figure is now converted into new tables.

11. Do not start a sentence with a number (line 351).

Response: The sentence has been re-write in revised manuscript. 

12. The ms has 71 references. Weed out excessive refs and restrict to 50 citations. Fig. 1 shows the world map for Pakistan is too basic. Remove this blue square fig.

Response: We have made efforts to reduce the number of references, and it now stands at 56. We added some new references in response to reviewer requests. 

The Fig. 1 has been revised

Reviewer #1: Comments:

1. The title of MS is still not clear.

Response: The MS title has been revised. 

2. The write-up of MS is still not standard.

For example: The write has been revised. 

Fat % = Fat weight (g)/weight if sample ×100

Response: The formula is that total fat extraction in fish samples is divided by the total weight of fish and then multiplied by 100

Fat Content (%) = (Weight of Fat in the Fish / Total Weight of the Fish) x 100

3. Before the harvesting of rice, rice stem characteristics were measured (Fig 3). This sentence can be of M & M part.

Response: The above sentence has been added to the revised manuscript

4. Crude protein % = Volume ×0.875/sample weight

Response: The crude protein is calculated by the Kjeldahl method, the revised formula is added in the manuscript as follows, 

Crude Protein (in %) = (N x 6.25) / W

Where:

N: Nitrogen content determined from the sample.

6.25: Conversion factor, assuming that proteins contain approximately 16% nitrogen.

W: The weight of the sample in grams.

5. Need references for the methods used in study for Yield parameter of rice

Response: The reference has been added to the rice yield parameter method.

6. Under different rice spacing densities, least square mean ± standard errors of crude protein

and fat of the muscles of the fish are presented in

Response: The above sentence has been revised.

7. Figures axis is invisible.

Response: The figure has been now changed into tabular form. 

Reviewer #2: 

1. The previous comments to the manuscript should have been incorporated in the resubmitted version. It was strongly suggested that the proof reading be done rigorously, but the current manuscript has many errors which could have been solved with proper proof reading. A few examples are as follows-

Response: Thank you for your feedback on our manuscript. We apologize for not incorporating previous comments and understand the need for rigorous proofreading. We have carefully addressed these issues in the revised version.

2. L20- “Genetically Male Tilapia” has already been mentioned to be referred to with “GMT”, yet L25, L27 mention “GMT (Tilapia)”

Response: The GMT (Tilapia) has been used in the entire manuscript. 

3. L26- “Delicious” is not the proper adjective to use for amino acids, in a scientific research article.

Response: The name is now changed to Umami Taste in the revised manuscript

4. L28- Full stop/Period between “acid categories” and “with RMD”

Response: The sentence has been revised. 

5. L45, L46- Use of “and” twice in the sentence.

Response: The sentence has been revised. 

6. L58- Use of “IAAF” and then “IAAFS” for Integrated Agri-Aquaculture Farming SystemL58- “IAAFS” is not required to be in the bracket.

Response: The word IAAFS has been changed.

7. L60- Grammatical error- “improves” instead of “improve”

Response: The word improve has been replaced with improves.

8. L66-L68- The whole sentence needs restructuring to remove grammatical errors

Response: The sentence has been revised as, the co-culture system has been observed to reduce pest communities and lower the requirement for agrochemical inputs, including pesticides and herbicides. 

9. L168- Both curly brackets and square brackets are used at once.

Response: The sentence has been revised. 

10. L152- “Meat analyses” is very unclear, the specific tissues to be analyzed are to be mentioned.

Response: The meat analysis has been revised as given below, After the collection of fishes removed the scales, skin, and inter muscular thorn. Fish muscles were cut and merged into a sample; these samples were carefully packaged in labeled, airtight plastic bags and stored at -20°C for further analysis 

11. The errors of the above-mentioned kind exist too frequently in the manuscript.

Response: We appreciated the feedback regarding the frequency of errors in the manuscript. We acknowledged the importance of addressing these issues to enhance the quality and credibility of our work. We thoroughly reviewed and revised the manuscript to rectify the mentioned errors and ensured its overall improvement.

1. The authors have stated that information on climate, elevation and temperature of the study area is essential to interpret the results of the study, yet there is no interpretation on that aspect in the discussion. There should also be a proper introduction for how these factors can be helpful in interpreting the results.

Response: The above information has been added to the introduction and discussion.

2. The experimental design states that, the study is conducted on 12 plots of 6500sq.ft each, fishes were stocked at a rate of 9000/ha, meaning approximately total 6500 fishes were stocked , of which only 3 fishes per plot were sampled for proximate and amino acid analysis? Were the fishes used in proximate analysis and Amino acid analysis the same? A sample size of three fish per plot is too small; the number of replicates should be more.

Response: we were stocked 9000/ha

Total sqft in one ha is 107693

Then calculate the total number of fish in a 6500sqft pond

=6500*9000/107693= 543.71 approximately 544

This means approximately 544 fish were stocked in a 6500sqft pond

12 experimental plots were used in this experiment. 

Three fishes per plot means 36 samples for proximate and 36 fishes for an amino acid profile in this way a total of 72 samples were collected for analysis. 

Proximate analysis: 3×12= 36 

Amino Acid Profile: 3×12= 36 

Total: 72

3. The experimental design is not clear.

Response: The experimental design has been clear in the revised manuscript

4. The authors have removed all information about zooplankton identification, but have responded to my previous query, hence I would like to mention that it is important to cite the taxonomic key and reference material used for zooplankton identification, as it not only helps the author to validate their work during the peer- review process, but also helps the reader to conduct similar study when the manuscript is published.

Response: The information has been added.

---

## [Decision Letter · Decision Letter 3]

14 Nov 2023

PONE-D-22-31911R3Optimizing Rice-Fish Co-Culture: Investigating the Impact of Rice Spacing density on Biochemical Profiles and Production of Genetically Modified Tilapia (Oreochromis spp.) and Cyprinus carpioPLOS ONE

Dear Dr. Inayat,

Thank you for submitting your manuscript to PLOS ONE. After careful consideration, we feel that it has merit but does not fully meet PLOS ONE’s publication criteria as it currently stands. Therefore, we invite you to submit a revised version of the manuscript that addresses the points raised during the review process.

We look forward to receiving your revised manuscript.

Kind regards,

SSS Sarma

Academic Editor

PLOS ONE

Journal Requirements:

**Additional Editor Comments:**

Dear authors

Your re-reviewed ms has come back with addition corrections. This is due to the fact that the previously two reviewers declined to take up your ms for further reviewing. As I see some potential in your ms, I have decided to give you another fair chance revise it further. Please note that your next revised ms (R4) will not be reivewed again. I will check it and send my observations to the Editor in Chief for his/her final decision. To increase the chances of positive consideration of your ms, please carefully revise your ms based on the observations of the two new reviewers. Plus please check the ms for grammatical, structural and/or typos.

Sincerely

Handling Editor

Reviewers' comments:

Reviewer's Responses to Questions

**Comments to the Author**

1. If the authors have adequately addressed your comments raised in a previous round of review and you feel that this manuscript is now acceptable for publication, you may indicate that here to bypass the “Comments to the Author” section, enter your conflict of interest statement in the “Confidential to Editor” section, and submit your "Accept" recommendation.

Reviewer #3: All comments have been addressed

Reviewer #4: (No Response)

2. Is the manuscript technically sound, and do the data support the conclusions?

Reviewer #3: Partly

Reviewer #4: Partly

3. Has the statistical analysis been performed appropriately and rigorously? 

Reviewer #3: N/A

Reviewer #4: Yes

4. Have the authors made all data underlying the findings in their manuscript fully available?

Reviewer #3: Yes

Reviewer #4: Yes

5. Is the manuscript presented in an intelligible fashion and written in standard English?

Reviewer #3: Yes

Reviewer #4: No

6. Review Comments to the Author

Reviewer #3: 1. The manuscript has some new ideas and/or data.

2. The authors are interested in rice production together with fish farming

3. The authors have considered the required variables of both these groups

4. Proximate composition of different elements has been considered.

The manuscript has some weak aspects which may be addressed:

1. Discuss the role of temperature on the differential fatty acid composition.

2. The authors showed higher crude protein and fat content in genetically male tilapia tan Cyprinus carpio. Since fat composition is influenced by temperature, authors are advised to discuss this aspect

3. Fish growth depends on plankton quality and quantity. Did the authors have any idea of these aspects in different treatments?

4. The fish species excrete considerable quantities of phosphorus and nitrogen compounds which influence the plankton dynamics.

5. Did you measure the N:P ratios during the study period.

Reviewer #4: Optimizing Rice-Fish Co-Culture: Investigating the Impact of Rice Spacing density onBioch emical Profiles and Production of Genetically Modified Tilapia (Oreochromis spp.) and Cyprinus carpio

In general, the idea of the manuscript is of relevance due sustainability purposes. The evaluation of the impact of rice spacing density on the production of GMT and Cyprinus carpio is understood, however, the ideas in some sections are hard to understand. Authors may consider rewriting and revising again the editing and translation of certain concepts. Also, there are a lot of errors on spaces and end points. General trends in results are lacking. The discussion needs to be the discussion must be strengthened and conclusive, the importance of the general aim must be highlighted. Some of the references used in introduction and discussion must be updated.

Abstract.

GMT- specify

Line 73. Aquaculture

Line 80. Herbicides. Several

Line 83. friendly. In

Line 89-90. Please revise the writhing

Line 93-95. The idea is not completely comprehensible

Line 96-98 unite ideas

Line 108. There are two points

Line 111-112 authors may consider eliminating the idea, as the aim is established further on the document. Perhaps authors can consider adding the hypothesis of the study.

Line 112 The study aims to evaluate the integration…

Line 130. Eliminate propose. For the present assessment…

Line 131-132 The idea is poorly understood

Is there any reference used for the experimental design?

Line 146 study. The acclimatization…

Line 148, 151, 157 double space

Line 167 the idea is vague, specifically the ending part of the idea.

References of the keys and manuals must be added in sample collection

Line 180 fish

Line 180-Previous to the experiment

Rewrite the idea

Is there any reference of the formula? Previous study were it has been used before?

Line 188 eliminate the

The monitoring of water quality is vague. Which parameters?

Line 232. One way anova?

Line 233-235 rewrite the idea

Line 278 are presented

Line 290 were presented?

In the section of results, the general idea must be established and described. Line 298-299 is out of line.

Describe Plankton community trends

Line 311- according to what?

Line 326- according to what authors or compared to previous studies?

No trends of the results are considered in the discussion as well as the use of tables or figures.

Line 341-342 the authors may consider mentioning why…

Line 359 elaborate idea

Fig 1 modify the map due to its poor quality

Fig 2 Experimental design considering 12 plots ()

Fig 3. Standard deviation of how many replicas?

Table 1 and 2 describe the treatments groups in ()

7. PLOS authors have the option to publish the peer review history of their article (what does this mean?). If published, this will include your full peer review and any attached files.

Reviewer #3: No

Reviewer #4: **Yes: **Brenda Karen González Pérez

---

## [Author Response · Author response to Decision Letter 3]

30 Nov 2023

RESPONSE LETTER

The, 

Editors, 

PLOS ONE

Subject: SUBMISSION OF REVISED PAPER 

Dear Editor,

Thank you for your email Dated: 15/Nov/2023, enclosing the reviewer’s comments. We appreciate the time and effort that you and the reviewers have dedicated to providing your valuable feedback on my manuscript. We are grateful to the reviewers for their insightful comments on my paper. We have carefully reviewed the comments and have revised the manuscript accordingly. Our responses are given in a point-by-point manner below. We hope the revised version is now suitable for publication and look forward to hearing from you in due course. 

You’re sincerely, 

Muhammad Inayat 

POINT-BY-POINT RESPONSE TO REVIEWER'S COMMENTS

Additional Editor Comments

Dear authors

Your re-reviewed ms has come back with addition corrections. This is due to the fact that the previously two reviewers declined to take up your ms for further reviewing. As I see some potential in your ms, I have decided to give you another fair chance revise it further. Please note that your next revised ms (R4) will not be reivewed again. I will check it and send my observations to the Editor in Chief for his/her final decision. To increase the chances of positive consideration of your ms, please carefully revise your ms based on the observations of the two new reviewers. Plus please check the ms for grammatical, structural and/or typos.

Response: Thank you for providing the opportunity for further revision. We appreciate your consideration and the potential you see in our manuscript. We carefully revised the manuscript based on their feedback. Additionally, we conducted a thorough review to rectify any grammatical, structural, or typographical issues.

Please use the space provided to explain your answers to the questions above. 

You may also include additional comments for the author, including concerns about dual publication, research ethics, or publication ethics. (Please upload your review as an attachment if it exceeds 20,000 characters)

Reviewer #3: 

1. The manuscript has some new ideas and/or data.

Response: Thank you for recognizing the manuscript's novel ideas and/or data; we appreciate your positive feedback.

2. The authors are interested in rice production together with fish farming

Response: Thank you for your positive feedback.

3. The authors have considered the required variables of both these groups

Response: Thank you for your positive feedback.

4. Proximate composition of different elements has been considered.

Response: We appreciate your recognition of the comprehensive consideration of proximate composition for different elements in our study.

The manuscript has some weak aspects which may be addressed: 

1. Discuss the role of temperature on the differential fatty acid composition.

Response: The role of temperature on the differential fatty acid composition has been added.

2. The authors showed higher crude protein and fat content in genetically male tilapia tan Cyprinus carpio. Since fat composition is influenced by temperature, authors are advised to discuss this aspect.

Response: Thank you for the insightful suggestion, the above detail has been added. 

3. Fish growth depends on plankton quality and quantity. Did the authors have any idea of these aspects in different treatments?

Response: In the rice-fish co-culture system, the author's detailed investigation of plankton quantity and quality under different treatments provided important insights into their potential effects on fish growth. The research highlights the significant role these factors play in determining the overall dynamics of the co-culture system as well as how different rice spacing densities affect phytoplankton and zooplankton ecosystems.

4. The fish species excrete considerable quantities of phosphorus and nitrogen compounds which influence the plankton dynamics.

Response: The research highlights the significant impact of fish species on the ecosystem by excreting significant quantities of phosphorus and nitrogen compounds. These excreted compounds play a pivotal role in affecting the dynamics of plankton within the system. By introducing additional nutrients, specifically phosphorus, and nitrogen, fish excretion becomes a key factor influencing the growth and composition of plankton communities. Understanding this aspect contributes to a more comprehensive understanding of the intricate relationships between fish, nutrients, and plankton dynamics in the studied ecosystem.

5. Did you measure the N:P ratios during the study period.

Response: No, I did not measure the N:P ratios during the study.

Reviewer #4: Optimizing Rice-Fish Co-Culture: Investigating the Impact of Rice Spacing density on Biochemical Profiles and Production of Genetically Modified Tilapia (Oreochromis spp.) and Cyprinus carpio

In general, the idea of the manuscript is of relevance due sustainability purposes. The evaluation of the impact of rice spacing density on the production of GMT and Cyprinus carpio is understood, however, the ideas in some sections are hard to understand. Authors may consider rewriting and revising again the editing and translation of certain concepts. Also, there are a lot of errors on spaces and endpoints. General trends in results are lacking. The discussion needs to be the discussion must be strengthened and conclusive, the importance of the general aim must be highlighted. Some of the references used in the introduction and discussion must be updated.

Response: We appreciate your feedback. We addressed clarity issues, revised editing and translation, corrected spacing and endpoint errors, included general trends in results, strengthened and concluded the discussion, and emphasized the importance of the general aim, Your input was valuable in refining the content.

Abstract.

GMT- specify

Response: The word GMT has been specified in the abstract 

Line 73. Aquaculture

Response: The aquaculture word has been revised. 

Line 80. Herbicides. Several

Response: The line has been revised. 

Line 83. friendly. In

Response: The line has been revised.

Line 89-90. Please revise the writhing

Response: The line has been rewritten in the revised manuscript.

Line 93-95. The idea is not completely comprehensible

Response: This has been comprehensibly explained in the revised manuscript. 

Line 96-98 unite ideas

Response: The ideas have been united in the revised manuscript. 

Line 108. There are two points

Response: The line has been revised.

Line 111-112 authors may consider eliminating the idea, as the aim is established further in the document. Perhaps authors can consider adding the hypothesis of the study. 

Response: These line has been updated.

Line 112 The study aims to evaluate the integration…

Response: The above information has been added.

Line 130. Eliminate propose. For the present assessment…

Response: The line has been revised.

Line 131-132 The idea is poorly understood

Response: These lines have been rewritten for better understanding. 

Is there any reference used for the experimental design?

Line 146 study. The acclimatization…

Response: The line has been revised in the revised manuscript. 

Line 148, 151, 157 double space

Response: Double space has been removed.

Line 167 the idea is vague, specifically the ending part of the idea.

Response: The idea has been updated. 

References of the keys and manuals must be added to sample collection

Line 180 fish

Response: The line has been updated.

Line 180-Previous to the experiment

Response: The line has been revised. 

Rewrite the idea

Is there any reference of the formula? Previous study were it has been used before?

Line 188 eliminate the

Response: The information has been revised.

The monitoring of water quality is vague. Which parameters?

Response: The water quality parameter has been added

Line 232. One way anova?

Response: The effects of different rice densities on the growth performance, meat proximate, and amino acid profile of two fish species were assessed through a factorial ANOVA using PROC GLM in SAS software (version 9.1). Both rice densities and fish species were treated as main effects, and their potential interactions were tested. This design allows for a comprehensive examination of the simultaneous influence of rice densities and fish species on the measured variables. Therefore, a factorial ANOVA was deemed appropriate for our study.

Line 233-235 rewrite the idea

Response: The idea has been re-written. 

Line 278 are presented

Response: The line has been updated

Line 290 were presented?

Response: The line has been revised.

In the section of results, the general idea must be established and described. Line 298-299 is out of line.

Response: The idea has been established and line 298-299 has been improved.

Describe Plankton community trends

Line 311- according to what?

Response: The information has been added.

Line 326- according to what authors or compared to previous studies?

Response: The information has been added.

No trends of the results are considered in the discussion as well as the use of tables or figures.

Line 341-342 the authors may consider mentioning why…

Response: The information has been revised.

Line 359 elaborates idea

Response: The idea has been elaborated.

Fig 1 modify the map due to its poor quality

Response: The map is modified. 

Fig 2 Experimental design considering 12 plots ()

Response: The information has been updated.

Fig 3. Standard deviation of how many replicates?

Response: In Fig 3. There are three replicates with each replicate

Table 1 and 2 describe the treatment groups in ()

Response: The information has been revised.

---

## [Editor Report · Decision Letter 4]

5 Dec 2023

Optimizing Rice-Fish Co-Culture: Investigating the Impact of Rice Spacing density on Biochemical Profiles and Production of Genetically Modified Tilapia (Oreochromis spp.) and Cyprinus carpio

PONE-D-22-31911R4

Dear Dr. Inayat,

We’re pleased to inform you that your manuscript has been judged scientifically suitable for publication and will be formally accepted for publication once it meets all outstanding technical requirements.

Kind regards,

SSS Sarma

Academic Editor

PLOS ONE

Additional Editor Comments (optional):

The final decision will be communicated to you by the Editorial Office.
---

## [Editor Report · Acceptance letter]

8 Dec 2023

PONE-D-22-31911R4 

Optimizing Rice-Fish Co-Culture: Investigating the Impact of Rice Spacing density on Biochemical Profiles and Production of Genetically Modified Tilapia (*Oreochromis spp.*) and *Cyprinus carpio*

Dear Dr. Inayat:

I'm pleased to inform you that your manuscript has been deemed suitable for publication in PLOS ONE. Congratulations! Your manuscript is now with our production department. 

Kind regards, 

on behalf of

Professor SSS Sarma 

Academic Editor

PLOS ONE